EMBO
reports

# Mitochondrial protein biogenesis in the synapse is supported by local translation

Bozena Kuzniewska[1,†] iD, Dominik Cysewski[2,†] iD, Michal Wasilewski[3,4] iD, Paulina Sakowska[5],
Jacek Milek[1], Tomasz M Kulinski[2,6] iD, Maciej Winiarski[7], Pawel Kozielewicz[3,5] iD, Ewelina Knapska[7],
Michal Dadlez[2], Agnieszka Chacinska[3,4,5] iD, Andrzej Dziembowski[2,6] iD &
Magdalena Dziembowska[1,*] iD

## Abstract

Synapses are the regions of the neuron that enable the transmission and propagation of action potentials on the cost of high energy consumption and elevated demand for mitochondrial ATP production. The rapid changes in local energetic requirements at dendritic spines imply the role of mitochondria in the maintenance of their homeostasis. Using global proteomic analysis supported with complementary experimental approaches, we show that an essential pool of mitochondrial proteins is locally produced at the synapse indicating that mitochondrial protein biogenesis takes place locally to maintain functional mitochondria in axons and dendrites. Furthermore, we show that stimulation of synaptoneurosomes induces the local synthesis of mitochondrial proteins that are transported to the mitochondria and incorporated into the protein supercomplexes of the respiratory chain. Importantly, in a mouse model of fragile X syndrome, *Fmr1* KO mice, a common disease associated with dysregulation of synaptic protein synthesis, we observed altered morphology and respiration rates of synaptic mitochondria. That indicates that the local production of mitochondrial proteins plays an essential role in synaptic functions.

**Keywords** fragile X syndrome; mitochondria; synapses; translation
**Subject Categories** Neuroscience; Organelles; Translation & Protein Quality

## Introduction

Synapses are spatialized zones of communication between neurons that enable the transmission and propagation of the signals.

Recently, it was shown that synapses are the regions of the neuron with the highest energy consumption. Thus, they have the highest demand for mitochondrial ATP production [1–3]. More specifically, it is the synaptic excitability that provokes temporal ion influx that will require millions of ATP molecules to be hydrolyzed to pump the ions back across the plasma membrane [3]. Maintaining resting potentials and firing action potentials is energetically expensive, as is neurotransmission on both the pre- and postsynaptic sides [4]. The rapid changes in local energetic demands at dendritic spines imply the role of mitochondria in the maintenance of their homeostasis.

Synapses underlay the phenomenon of the plastic change referred to as synaptic plasticity. Some forms of synaptic plasticity require mRNA translation in the postsynaptic region [5,6]. This process proved to be extremely important for the physiology of neurons, and its dysfunction leads to abnormalities observed in the disease syndromes such as fragile X syndrome (FXS, a mutation in fragile X mental retardation 1 gene, *FMR1*) and autism [7]. The discovery of actively translating polyribosomes in dendritic spines raised the question of which mRNAs are locally translated and what their function is [8]. Since then, many experimental approaches have been applied to identify mRNAs transported to the dendrites and translated upon synaptic stimulation. Transcriptomic studies suggested that a large portion of proteins present in dendrites and at dendritic spines can be synthesized locally on the base of mRNA specifically transported to this compartment [9]. These results reveal a previously unappreciated enormous potential for the local protein synthesis machinery to supply, maintain, and modify the dendritic and synaptic proteome. However, one of the critical questions in this field that remain uncovered is the global impact of local translation on synaptic functions. Recently, the role of mitochondria in the plasticity of dendritic spines was revealed on the postsynapse [10]

1 Laboratory of Molecular Basis of Synaptic Plasticity, Centre of New Technologies, University of Warsaw, Warsaw, Poland
2 Institute of Biochemistry and Biophysics, PAS, Warsaw, Poland
3 Laboratory of Mitochondrial Biogenesis, Centre of New Technologies, University of Warsaw, Warsaw, Poland
4 ReMedy International Research Agenda Unit, University of Warsaw, Warsaw, Poland
5 Laboratory of Mitochondrial Biogenesis, International Institute of Molecular and Cell Biology, Warsaw, Poland
6 Laboratory of RNA Biology, International Institute of Molecular and Cell Biology in Warsaw, Warsaw, Poland
7 Nencki Institute of Experimental Biology, Warsaw, Poland
*Corresponding author. Tel: +48 22 55 43721; E-mail: m.dziembowska@cent.uw.edu.pl
†These authors contributed equally to this work

and in axons where the late endosomes were shown to serve as a platform for local translation [11].

Herein, we employed quantitative mass spectrometry and *in vitro* stimulation of isolated mouse synapses (synaptoneurosomes) to create a comprehensive view of local protein synthesis in neurons. Strikingly, the third most numerous group of proteins synthesized in the synapses represented ones imported into the mitochondria. The proteomic data were further supported by the sequencing of mRNAs bound with actively translating polyribosomes. Our results show that an essential pool of mitochondrial proteins is locally produced at the synapse, indicating that mitochondrial biogenesis takes place locally to maintain the functional mitochondria in axons and dendrites. We further show that stimulation of synaptoneurosomes induces the local synthesis of mitochondrial proteins that are transported to the mitochondria and incorporated into the respiratory chain complexes. That contributes to mitochondrial biogenesis in neurons, and a logical consequence of this fact would be a dysregulation of mitochondrial function in the conditions that deal with dysregulated synaptic translation, such as FXS. Consequently, we have shown mitochondrial dysfunction in the *Fmr1*KO mice synapses.

## Results and Discussion

### Mitochondrial proteins represent a significant fraction of locally synthesized proteins in synaptoneurosomes

To explore the dynamics of synaptic proteome upon neuronal stimulation, we employed a simple model system, isolated synaptoneurosomes (SN). Synaptoneurosomes are a fraction of brain homogenate obtained by the series of filtrations and centrifugations. The resulting fraction is enriched in synapses containing both pre- and postsynaptic compartments (Fig 1A). The Western blot on fractions obtained during SN preparation shows enrichment of pre- and postsynaptic markers (Psd95, GluA1, GluA2, Nlgn3, and synaptophysin) as well as depletion of cytosolic markers (Gapdh and Hsp90) in synaptoneurosomes as compared to the homogenate. Nuclear markers (Kdm1 and c-Jun) are barely detectable in synaptoneurosomal fraction. Glia marker (Gfap) is present in synaptoneurosomes; however, it is not enriched (Fig 1B). Directly after the isolation, synaptoneurosomes can be *in vitro* stimulated to induce local protein translation. We have chosen the stimulation protocol that promotes the induction of N-methyl-D-aspartate receptors (NMDA-Rs) on the synaptoneurosomes, which initiate calcium signaling in neurons and physiological conditions in the brain, and leads to long-lasting responses such as long-term potentiation (LTP) [12]. For this, we treated synaptoneurosomes for 30 s with NMDA and glutamate and added a selective NMDA-R antagonist (APV) to avoid the induction of excitotoxicity. This treatment produces the transient phosphorylation of extracellular signal-regulated protein kinases (ERKs) in synaptoneurosomes, which reflects activity-induced calcium influx mediated by NMDA receptors (Fig EV1) [13,14]. Next, in order to study activity-induced protein translation, we incubated synaptoneurosomes with radioactive methionine/cysteine prior to NMDA-R stimulation. We observed the incorporation of radioactive amino acids into the newly synthesized proteins at 15, 30, 60, and 120 min, as revealed by the autoradiography of

the SDS–PAGE gel (Fig 1C). In the control experiments, when synaptoneurosomes were pretreated with cytoplasmic protein synthesis inhibitors, such as puromycin (Fig 1C), cycloheximide, or anisomycin (Fig EV1C), significant inhibition of the translation visualized by $^{35}$S-Met/Cys incorporation was observed. This effect was not observed with chloramphenicol, an inhibitor of mitochondrial translation (Fig 1C). The residual staining is caused by non-specific interactions of radiolabeled amino acids with proteins as verified by incubation of $^{35}$S-Met/Cys with inactivated synaptoneurosomes (Fig EV1B).

Protein translation in isolated synaptoneurosomes depleted of nuclei and cell bodies relies on the pool of mRNAs transported into the synapses in the brain. Therefore, we reasoned that all proteins whose abundance will increase after the *in vitro* stimulation would be locally synthesized on the base of these mRNAs. Thus, in order to identify the proteins newly synthesized in stimulated synaptoneurosomes, we performed mass spectrometry (label-free quantification—LFQ) and compared the abundance of proteins in unstimulated and NMDA-R-stimulated SN (Fig 1D). The total number of identified proteins was 2200 (FDR 1%). To gain some insight into the biological functions of identified proteins, we have performed standard gene annotation using DAVID. Strikingly, the third most numerous group was annotated as proteins imported to the mitochondria after the nucleotide binding and synaptic proteins (Fig 1E). 24% of all upregulated proteins were mitochondrial (Fig 1F).

Because of the small number of proteins that reached statistical significance in the LFQ analysis, we turned into the quantitative mass spectrometry based on isobaric labeling techniques and we performed labeling of the samples with iTRAQ or TMT tag (Fig 2A and B). Quantitative mass spectrometry heavily depends on the physicochemical properties of peptides and dynamic range of samples, which limits the depth of analysis. Some of the tryptic peptides would be better detectable unmodified, other when modified for example, with stable isobaric labeling techniques such as iTRAQ or TMT tag. Therefore, in order to gain a more in-depth insight into the nature of synaptic proteome and overcome physicochemical limitations, we measured the same samples using two quantitative methods: iTRAQ8 and TMT10 on the same biological material. We identified 1,942 proteins using iTRAQ8-plex and 2,508 in TMT10-plex. Altogether in three independent proteomic approaches, we identified 3,201 proteins out of which we were able to quantify 2,080. 516 of those proteins were significantly upregulated in the stimulated synaptoneurosomal samples (Mann–Whitney $P < 0.05$). The list of identified proteins is attached as Dataset EV1. Next, we performed gene annotation enrichment analysis with DAVID on proteins identified as significantly upregulated after NMDA-R stimulation (assembled from all three proteomic analyses). Here, we observed mitochondrial proteins to be the second most abundant category of identified proteins (Fig 2C).

To look more specifically on mitochondrial proteins, we equated our results to a mitochondrial database of MitoCarta 2.0. Mitochondrial proteins constituted 19% (393) of all quantified proteins and 13.4% (69) of upregulated ones (Fig 2C). This finding is remarkable since mitochondrial proteins (1,160 gene names, MitoCarta 2.0) represent only 5% of all protein sequences in the database downloaded from UniProt (52539 entries) and used for the results annotation.

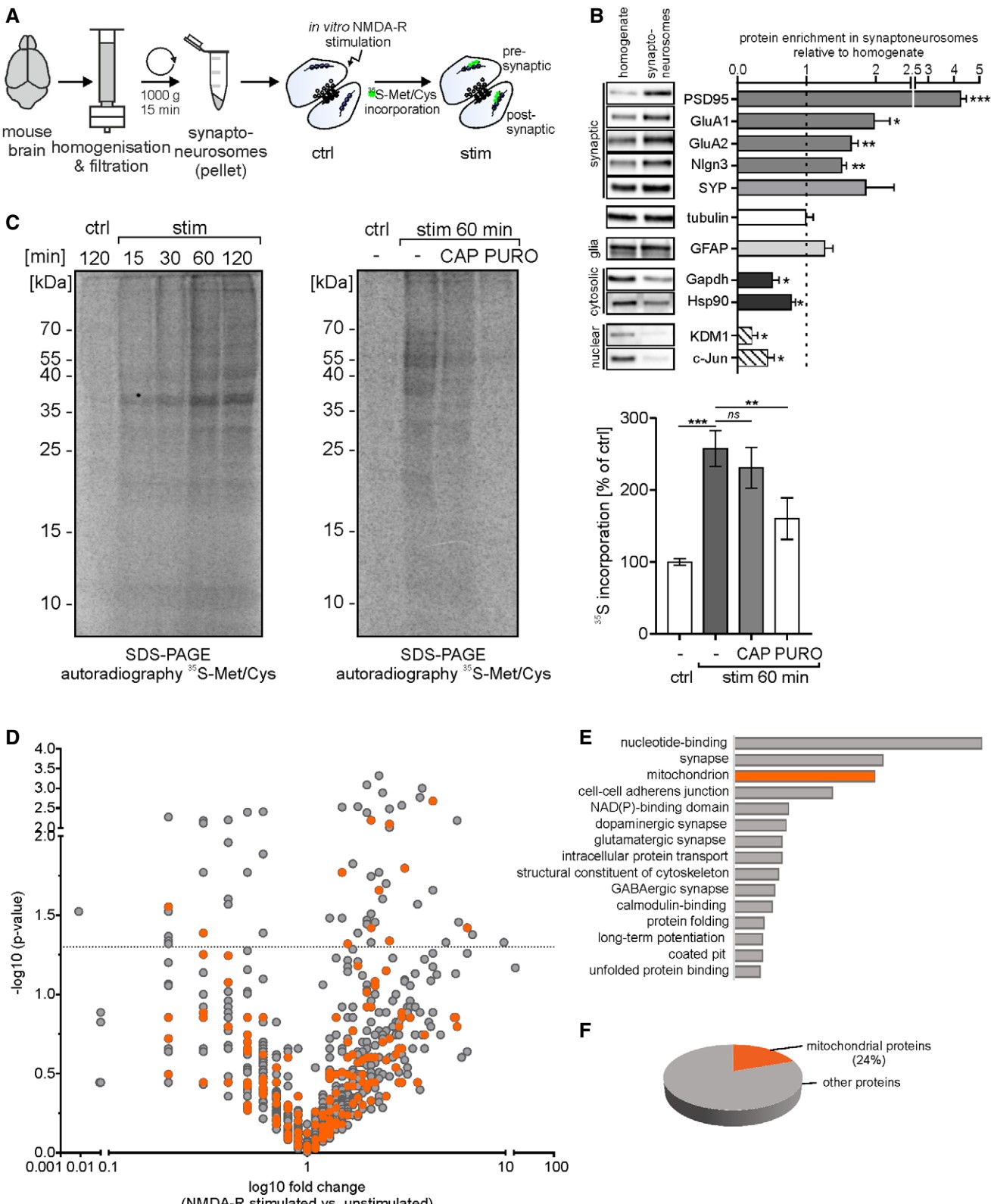

**Figure 1.**

**Figure 1.  Mitochondrial proteins represent a significant fraction of locally synthesized proteins in synaptoneurosomes.**

A   Workflow of the experiment presented in panel (C), depicting synaptoneurosome (SN) isolation and *in vitro* stimulation in the presence of radioactive [35]S-labeled methionine/cysteine mix.

B   Western blot on fractions obtained during SN preparation revealed the enrichment of both pre- and postsynaptic markers, but not glia marker (GFAP) in the SN fraction. Cytosolic markers were depleted, while nuclear markers were barely detected in the SN ($n = 4$ biological replicates, ***$P < 0.001$, **$P < 0.01$, *$P < 0.05$; one sample $t$-test; error bars indicate SEM).

C   Synaptoneurosomes were NMDA-R-stimulated and incubated with radioactive [35]S-methionine/cysteine mix for 15–120 min. After labeling, proteins were separated on SDS–PAGE and newly synthesized proteins in SN labeled with [[35]S] were visualized by autoradiography (left panel). To determine the contribution of mitochondrial and cytosolic translation, the labeling was preceded by treatment with chloramphenicol (CAP, 50 μg/ml) or with puromycin (PURO, 3 mM), respectively (right panel). After 1 h after the stimulation, a significant increase in [35]S-labeled proteins was observed ($n = 6$ biological replicates, ***$P < 0.001$; repeated-measures one-way ANOVA, *post hoc* Sidak's multiple comparisons test) and the CAP treatment did not affect the overall levels of *de novo* synthetized proteins ($n = 6$ biological replicates, ns; $P = 0.220$). In contrast, PURO treatment significantly inhibited *de novo* protein synthesis in synaptoneurosomes ($n = 6$ biological replicates, **$P = 0.0091$, repeated-measures one-way ANOVA, *post hoc* Sidak's multiple comparisons test). Error bars indicate SEM.

D–F   Results of mass spectrometry (label-free quantification—LFQ) analysis identifying proteins with increased abundance in NMDA-R-stimulated (20 min) synaptoneurosomes. (D) Volcano plot (Mann–Whitney test, $n = 4$ per group) showing abundance of identified proteins in stimulated synaptoneurosomes as compared to unstimulated. Mitochondrial proteins are depicted in orange. The vertical line defines the $P$-value statistical significance cutoff. (E) Biological functions of identified proteins annotated using DAVID. Third top functional category identified as mitochondrial ones. (F) Pie chart showing percentage share of mitochondrial proteins among all identified upregulated proteins.

Proteomic mass spectrometry data from Dataset EV1 and the software program Cytoscape (version 3.5.1) were used to generate a graphic representation of known protein–protein interactions for proteins significantly upregulated by NMDA-R stimulation (Fig 2D).

The number of quantified proteins (3201) could be considered as relatively low for shotgun MS experiment; however, one should take into account the specificity of the synaptoneurosomal preparations depleted of nuclear and probably many cytoplasmic proteins. Importantly, 516 proteins upregulated in synaptoneurosomes by the stimulation correspond to as much as 20% of all identified proteins. The average fold change was relatively low, which was, however, expected because of the translational capacity of synaptic ribosomes.

To strengthen the proteomic data, we additionally analyzed synaptoneurosomal transcriptome engaged in translation. For this, we took advantage of the fact that local synthesis of synaptic proteins is performed on-site with the pool of available ribosomes [15] that can be isolated from synaptoneurosomes and fractionated on the sucrose gradient according to their molecular weight [16]. Thus, we fractionated polyribosomes from stimulated synaptoneurosomes to identify the mRNAs associated with polyribosomal fractions in response to the stimulation (Fig 2E). Then, we performed RNA-seq experiments in which we detected 17,160 different transcripts, 3,078 (18%) of which were detected at the protein level. Then, we calculated the abundance of given mRNA in the polyribosomal fraction in response to NMDA-R stimulation comparing to total mRNA from the unstimulated sample which is visualized as an MA plot (Fig 2F). Importantly, both transcripts encoding all upregulated proteins (green dots) and mitochondrial proteins (orange dots) detected by the mass spectrometry represented highly abundant mRNAs with good ribosome association values. They were generally enriched in the polyribosomal fractions as visualized on the density curves (Fig 2F). Quantitatively, 64% of all upregulated proteins were also enriched on the mRNA level supporting their activity-induced polyribosomal binding and translation. In the case of the remaining 36% of proteins, we did not detect their enrichment in the polyribosomal fraction. Nevertheless, they were mostly already present in the polysomal fraction. Thus, although they have the potential to be translated locally, the mechanism of regulation remains to be established. Importantly, 56 of 69 upregulated

mitochondrial proteins were also overrepresented at the mRNA level in the polyribosomal fraction (Fig 2F).

Next, we verified RNA-seq data using qRT–PCR on the polysomal fractions isolated from synaptoneurosomes (selected mRNAs are listed in Fig EV2). Indeed, we could see the transcripts encoding for mitochondrial proteins shift toward the polyribosomal fractions upon the NMDA-R stimulation (Fig EV2).

## Locally synthesized proteins build mitochondria in the synapse

The large fraction of locally synthesized mitochondrial proteins identified in our mass spectrometry analysis were the constituents of the respiratory chain complexes (see Dataset EV1). Mitochondria in presynaptic terminals of synapses were for a long time used as a hallmark of synapses, and their real role in neurotransmission is recently being determined [17]. The studies of axonal local translation in squid giant axon and presynaptic nerve terminals of the photoreceptor neurons revealed the presence of mRNA population containing nuclear-encoded mitochondrial proteins [18,19]. Synapses in the brain contain mitochondria; therefore, to confirm that the mitochondria located in SN are functional, we measured $O_2$ consumption following supplementation in respiratory substrates, uncouplers, and inhibitors. Quantitative respirometry was performed in parallel to compare the respiration rates between permeabilized HEK293T cells and SN. We detected similar responses to the addition of respiratory substrates for complex I (malate and pyruvate) and complex II (succinate) in the case of HEK293T cells and SN pointing at their similar respiration status (Fig 3A). We also compared the native mitochondrial protein complexes from SN and HEK293T cells obtained by digitonin extraction and blue-native gel electrophoresis (BN-PAGE), the golden standard for visualization of respiratory complexes and supercomplexes present in mitochondria (Fig 3B). The characteristic pattern of bands representing the native mitochondrial respiratory chain complexes appeared to be similar in SN and HEK293T cells although more high molecular weight bands in the region of supercomplexes were revealed in SN. Next, the BN-PAGE gels were electrotransferred onto the PVDF membrane for Western blot analysis. We used the panel of antibodies specifically recognizing the protein components of complex I (Ndufb8), complex II (Sdha), complex III

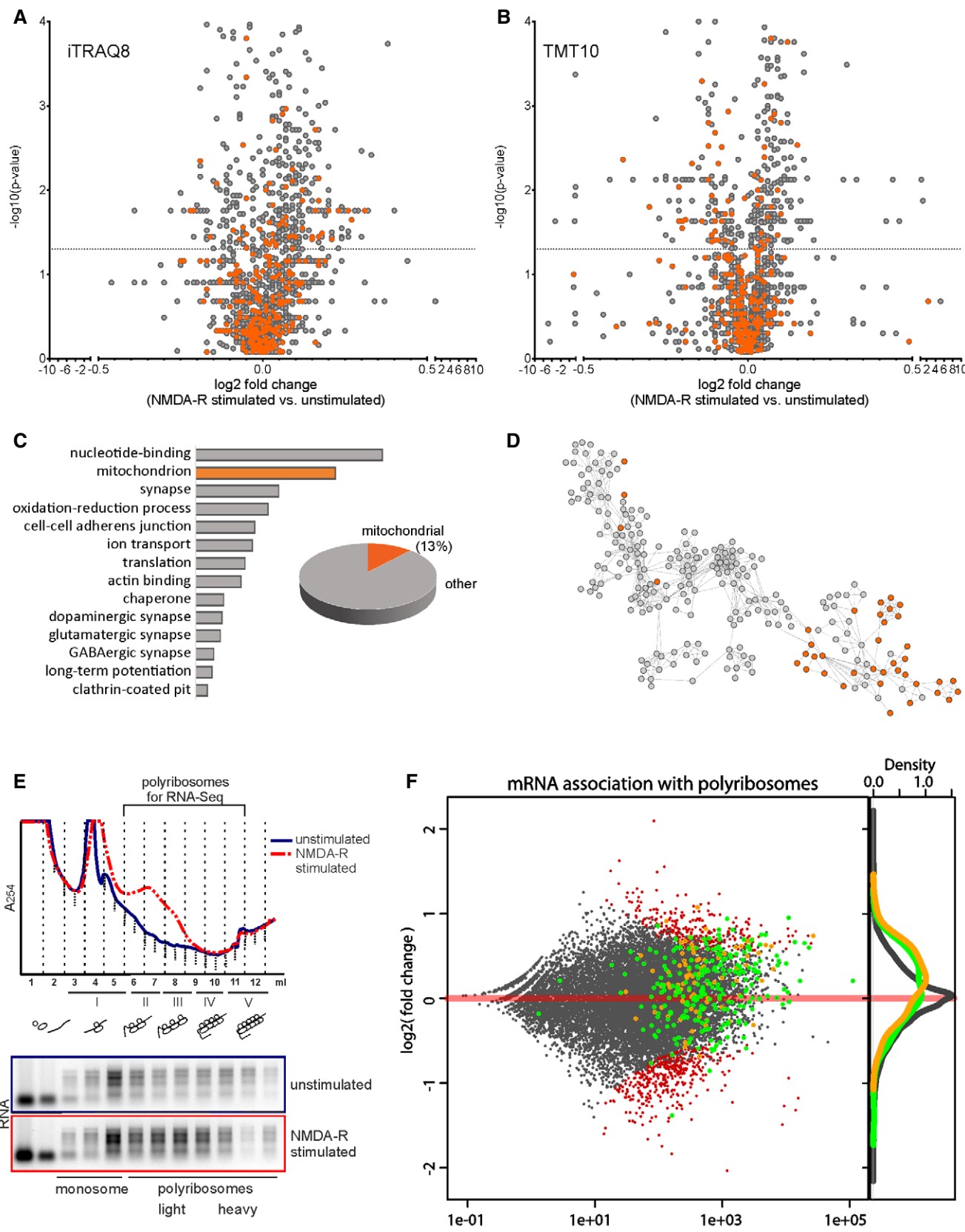

**Figure 2.**

**Figure 2.** **Identification of proteins locally synthesized in synaptoneurosomes using high-resolution quantitative mass spectrometry and RNA sequencing.**

A–D  Results of high-resolution quantitative mass spectrometry analysis identifying proteins with increased abundance in stimulated synaptoneurosomes. Volcano plots (Mann–Whitney test, Benjamini–Hochberg correction, $n = 5$ per group) showing abundance of identified proteins in NMDA-R-stimulated (20 min) synaptoneurosomes as compared to unstimulated using iTRAQ8-plex (A) and TMT10-plex (B) labeling methods. Mitochondrial proteins are depicted in orange. The vertical line defines the $P$-value statistical significance cutoff. For nine mitochondrial proteins, their mRNA abundance on polyribosomes was quantified; the data are presented in Fig EV2. (C) Biological functions of proteins identified as significantly upregulated after NMDA-R stimulation (assembled from all three proteomic analyses: LFQ, iTRAQ8, and TMT10) annotated using DAVID. Second top functional category identified as mitochondrial ones. Mitochondrial proteins constituted 13% (69) of all upregulated proteins. (D) Interactome of identified proteins, mitochondrial proteins shown in orange.

E  RNA absorbance profiles from polyribosomal fractionation of unstimulated and NMDA-R-stimulated (20 min) synaptoneurosomes. Upon NMDA-R stimulation, a fraction of mRNAs is shifted toward heavy polyribosomal fraction, which represents actively translating ribosomes. The fraction of polyribosomes used for RNA-seq experiment is depicted.

F  Comparison of proteomic data with RNA-seq experiments on mRNAs associated with polyribosomes from synaptoneurosomal samples. Ribosome association of synaptoneurosomal mRNAs determined by normalizing RNA-seq values of polyribosome-associated mRNA with total mRNA ($y$-axis). Green dots represent mRNAs encoding proteins upregulated upon stimulation (identified in MS experiments), while orange dots represent mRNAs encoding mitochondrial proteins. Density curves for transcripts encoding mitochondrial proteins (orange) and upregulated proteins (green) detected by the mass spectrometry are present on the right side of the plot. Please note that both classes represent high abundant mRNAs with good ribosome association values. Transcripts with statistically significant increased or decreased abundance on the polyribosomes in response to the stimulation are depicted in red.

(Uqcrc2), complex IV (Cox6a), and complex V (Atp5a) that revealed the presence of these complexes in the BN-PAGE gels (Fig 3C). The pattern and identity of proteins were in agreement with a well-established profile of the native respiratory chain complexes from mouse heart [20]. Interestingly, we observed more bands corresponding to supercomplexes in SN than in HEK293T, which confirmed the Coomassie staining (Fig 3B and C).

For the undeniable characterization of the protein complexes visible on the BN-PAGE gels, we dissected eight bands from the polyacrylamide gel and analyzed them by mass spectroscopy (Fig 3D). The protein components of mitochondrial respiratory chain complexes were identified (listed in Table EV1).

To establish that locally translated mitochondrial proteins are effectively transported into the mitochondria, we loaded synaptoneurosomes with radioactive methionine/cysteine, stimulated NMDA-Rs allowing local protein synthesis, and assayed for incorporation of radioactive proteins into the native mitochondrial respiratory chain complexes on BN-PAGE (Fig 4A). The incorporation of newly synthesized radiolabeled proteins into complexes detected by BN-PAGE resulted in a pattern overlapping with the mature complexes detected by the Coomassie staining (Fig 3B). Additional bands formed by radiolabeled proteins likely represented import and complex assembly intermediates [21]. As the control, the SN were treated with CCCP or VOA to abolish mitochondrial electrochemical potential and thus inhibit protein transport into the mitochondria. This treatment effectively inhibited the activity-induced appearance of radioactive respiratory chain complexes containing newly synthesized proteins in the mitochondria (Fig 4B and C). The inhibition was also observed when SNs were pretreated with protein translation inhibitor puromycin (Fig 4C). Altogether, we established that synaptically translated proteins are imported into mitochondria and assembled into the functional respiratory chain complexes.

### Altered mitochondrial functions and morphology in synapses of *Fmr1* KO mice, a model of a fragile X syndrome

Having established that mitochondrial proteins are indeed synthesized locally and incorporated into respiratory chain complexes in synaptic mitochondria, we asked a question about the physiological consequences of this phenomenon. One of the conditions in which

the local synaptic translation is genetically dysregulated is a human neurodevelopmental disorder known as fragile X syndrome. We took advantage of the fact that in the mouse model of this disease, an *Fmr1* KO mouse, synaptic protein synthesis is also dysregulated. Namely, in the basal state synaptic translation is upregulated and poorly responds to stimulation [7].

We aimed at the analysis of mitochondrial metabolism, respiration, and morphology in *Fmr1* KO synapses (Fig 5A).To compare mitochondrial metabolism in SN isolated from *Fmr1* KO and WT mice, we assayed the rates of electron flow through the mitochondrial electron transport chain following reduction of tetrazolium redox dye which receives electrons from cytochrome $c$. We used a set of 30 different metabolic substrates such as L-malate, succinate, D,L-$\alpha$-glycerol-PO$_4$, and D,L-$\beta$-hydroxy-butyric acid. (MitoPlates, Biolog). We observed significantly higher utilization of succinic acid by mitochondria from *Fmr1* KO mice (Fig 5B). A similar trend was observed in case of several other substrates such as D,L-$\alpha$-glycerol-PO$_4$ and D,L-$\beta$-hydroxy-butyric acid although in these cases the increase in *Fmr1* KO mitochondria failed to reach significance. Mitochondrial content in *Fmr1* KO and WT synaptoneurosomes was assessed by measuring citrate synthase enzymatic activity (Fig EV3B) as well as mitochondrial DNA levels using qRT–PCR. We did not detect any significant differences in mitochondrial content in *Fmr1* KO and WT SN (Fig EV3C).

Next, to study the respiration of synaptic mitochondria more precisely, we used a high-resolution respirometry technique that allows measuring O$_2$ consumption in SN isolated from WT and *Fmr1* KO mice (Fig 5C). Similarly to the tetrazolium dye method, we found that in the presence of succinic acid, which provide electrons to complex II, the SN from *Fmr1* KO mice respired at significantly higher rate than the control. Moreover, respiration of SN from *Fmr1* KO mice was increased in the presence of ascorbate and TMPD that provide electrons directly to complex IV. In line with this observation, in the Drosophila model of FXS, *dfmr1*, mitochondria have significantly increased maximum electron transport system (ETS) capacity similar to what we observed in *Fmr1* KO synapses [22].

Mitochondria are highly dynamic organelles, and local protein synthesis may affect not only their biochemical properties but also morphology, movements, and fusions. Thus, we studied the morphology of the mitochondria in the brains of the *Fmr1* KO

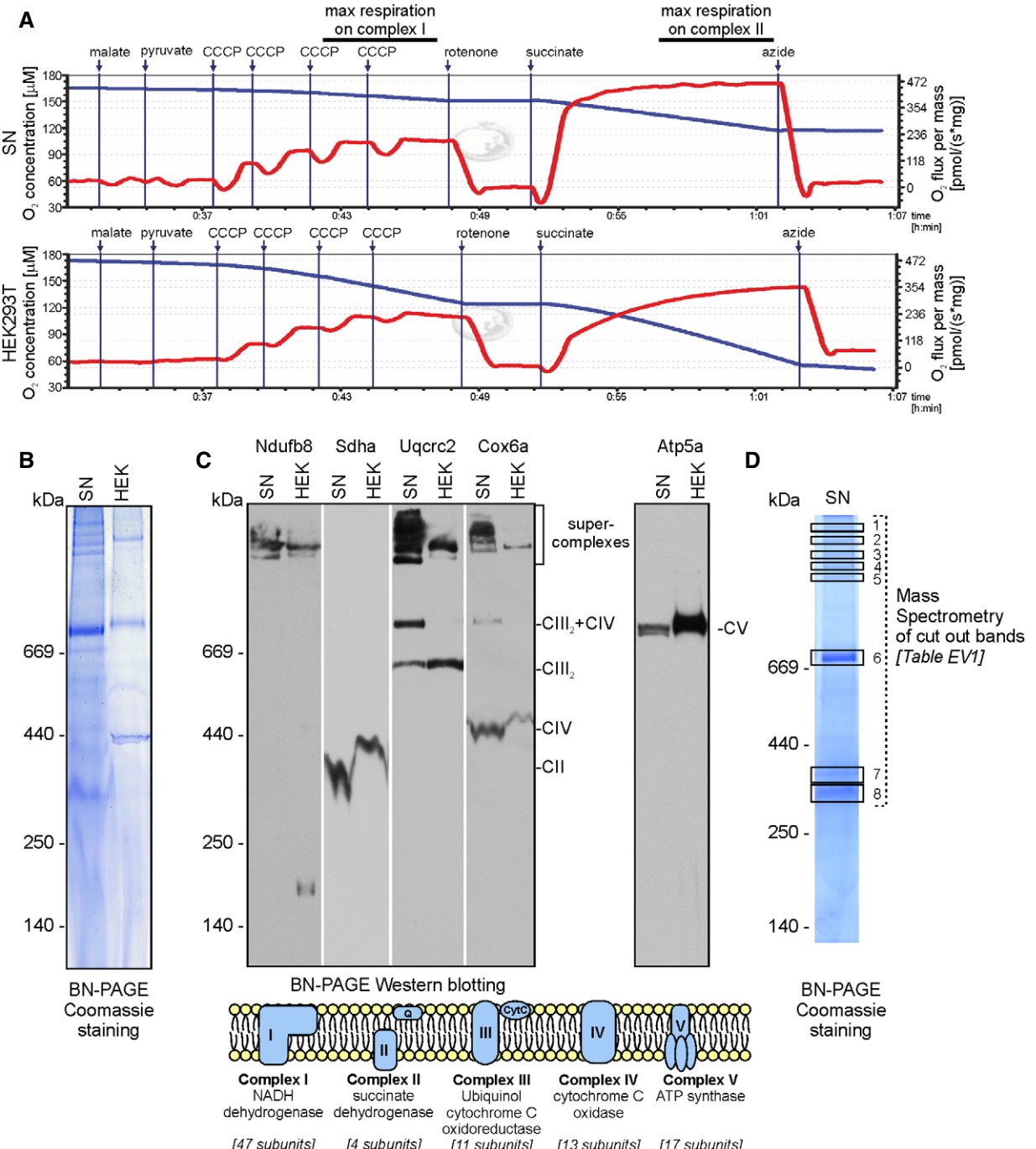

**Figure 3. Characterization of mitochondrial respiratory complexes and their activity in synaptoneurosomes.**

A   Respiration activity of permeabilized synaptoneurosomes (upper panel) as compared to permeabilized HEK293T (lower panel). Oxygen concentration (blue line) and oxygen consumption (red line) measured using O2k Oxygraph are shown.

B   Blue native gel electrophoresis (BN-PAGE) analysis of mitochondrial respiratory chain complexes from synaptoneurosomes as compared to HEK293T mitochondria.

C   Detection of respiratory chain complexes and supercomplexes in SN and HEK293T mitochondria using BN-PAGE followed by Western blot using antibodies specific for: complex I (Ndufb8), complex II (Sdha), complex III (Uqcrc2), complex IV (Cox6a), and complex V (Atp5a). Note similar composition of OXPHOS supercomplexes and complexes in synaptoneurosomes as in cellular mitochondria.

D   Selected bands (1-8) visible in blue native gel were excised and analyzed using mass spectrometry (LC-MS). Proteomic analysis confirms the presence of protein subunits of different respiratory chain complexes in indicated bands. See Table EV1 for identified peptides.

mice using electron microscopy, which revealed prominent morphological alterations of the mitochondria at the synapses of *Fmr1* KO neurons. In the synapses of WT mice, we observed morphologically normal mitochondria with a regular distribution of cristae and matrix of typical electron density (Fig 5D). In contrast, striking morphological abnormalities were observed in mitochondria of *Fmr1* KO synapses, both in dendritic spines

(postsynaptic side) and in the presynaptic terminals (Fig 5D). Mitochondrial pathology at the synapses consisted of substantial changes in the shape and size, alteration in the arrangement of the cristae, and in some cases accumulation of osmiophilic material (Fig EV3D). Generally, we observed small, round mitochondria with irregular cristae and a very high electron density of the matrix.

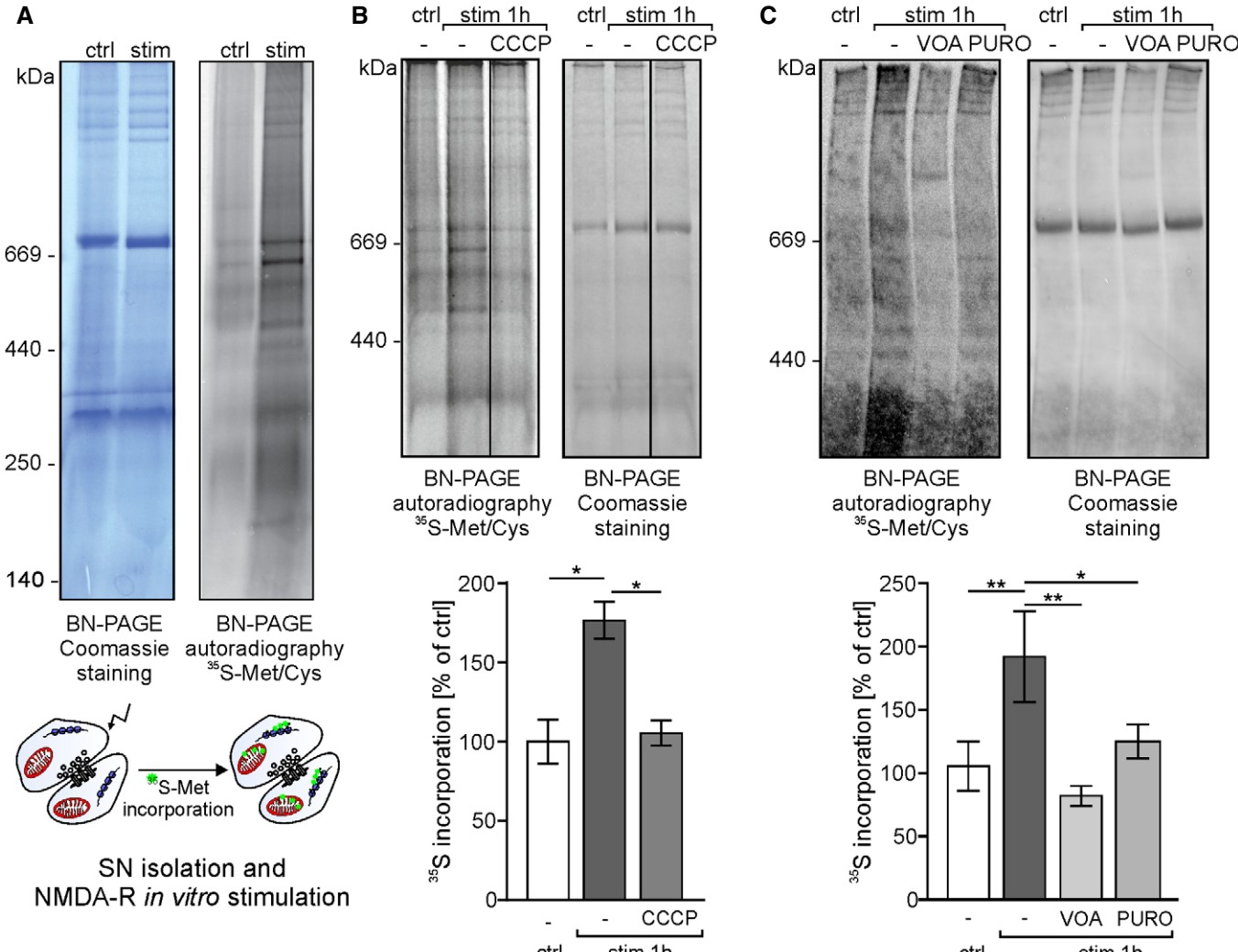

**Figure 4. Mitochondrial proteins locally synthesized in the synapse are incorporated into respiratory chain complexes.**

A   Synaptoneurosomes were NMDA-R-stimulated and incubated with radioactive $^{35}$S-methionine/cysteine mix for 1 h. Samples were separated using blue-native electrophoresis (BN-PAGE). Protein complexes were visualized by Coomassie staining (left) and autoradiography (right). Autoradiography of the BN-PAGE gel shows *de novo* synthetized mitochondrial proteins that are incorporated into respiratory complexes.

B, C   To block protein import and incorporation of newly synthesized proteins into respiratory complexes, the mitochondrial electrochemical potential was abolished by treatment with carbonyl cyanide m-chlorophenyl hydrazone (10 μM CCCP) (B) or VOA mixture (containing 1 μM valinomycin, 20 μM oligomycin, and 8 μM antimycin) (C). Additionally, in order to block protein synthesis, SN were pretreated with puromycin (PURO, 3 mM) (C). Protein complexes were separated on BN-PAGE and visualized by Coomassie staining and autoradiography. One hour after the stimulation, significant increase in $^{35}$S-labeled proteins was observed (B, *n* = 4 biological replicates, \**P* < 0.0145; C, *n* = 5 biological replicates, \*\**P* = 0.0023; repeated-measures one-way ANOVA, *post hoc* Sidak's multiple comparisons test). Blocking protein import into the mitochondria significantly inhibited the incorporation of $^{35}$S-methionine into respiratory chain complexes when samples were incubated in the presence of CCCP (B; *n* = 4 biological replicates, \**P* = 0.02, repeated-measures one-way ANOVA, *post hoc* Sidak's multiple comparisons test) or VOA (C; *n* = 5 biological replicates, \*\**P* = 0.0044, repeated-measures one-way ANOVA, *post hoc* Sidak's multiple comparisons test). Also, significant inhibition of $^{35}$S-methionine/cysteine incorporation into mitochondrial protein complexes in synaptoneurosomes was observed when samples were stimulated in the presence of puromycin (C, *n* = 5 biological replicates, \**P* = 0.0236, repeated-measures one-way ANOVA, *post hoc* Sidak's multiple comparisons test). Error bars indicate SEM.

Source data are available online for this figure.

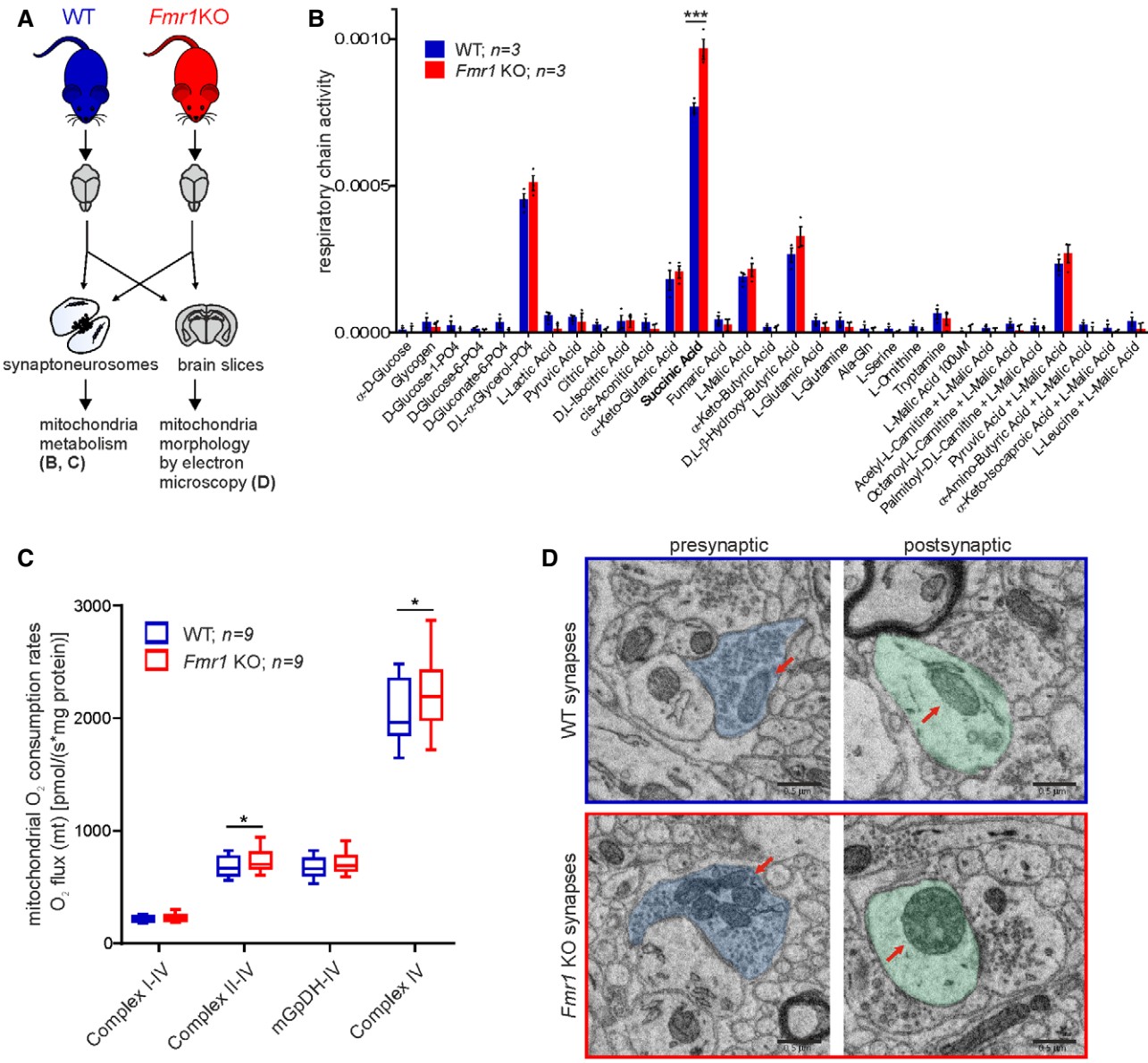

**Figure 5. Altered mitochondrial respiration and morphology in *Fmr1* KO synapses.**

A   *Fmr1* KO mice were used as a mouse model of dysregulated local translation. Mitochondria morphology in the brain slices and mitochondria metabolism in synaptoneurosomes were assessed in *Fmr1* KO and WT mice.

B   Equal amounts of synaptoneurosomal samples isolated from mouse brains (*Fmr1* KO and wild-type littermates) were permeabilized by saponin and assayed on MitoPlates for 2 h. Results are presented as the average rate/min/μg of protein, +/- SEM (n = 3 per genotype; two-way ANOVA, *post hoc* Sidak's multiple comparisons test; ***P < 0.0001).

C   Mitochondrial $O_2$ consumption rates in SN were measured by high-resolution respirometry in the presence of following respiratory substrates: malate, pyruvate, and glutamate for complexes I-IV, succinate for complexes II-IV, glycerol 3-phosphate for mGpDH-IV, and ascorbate and TMPD for complex IV, and normalized to protein content. *Fmr1* KO mitochondria show elevated levels of respiration in the presence of succinate as well as ascorbate and TMPD (*P = 0.0483 and *P = 0.0360, respectively, n = 9 per genotype, paired two-tailed *t*-test). Data are presented as a box-and-whiskers graph (the box extends from 25th to 75th percentiles, central horizontal line is plotted at the median, and whiskers show 5th – 95th percentile).

D   Ultrastructure of *Fmr1* KO brain sections revealed important differences in mitochondria morphology when compared to wild-types. Mitochondria are indicated by arrows; presynaptic part in *blue*, postsynaptic part in *green*. Three mice per genotype were analyzed. Scale bars, 0.5 μm. See also Fig EV3.

Moreover, we found mitochondria with partially impaired structure, discontinuous, incomplete outer membrane, and altered arrangement of the cristae. Occasionally, some of the abnormal mitochondria were swollen with pale matrix (reduced matrix density) without or with very few cristae. Also, the presence of

small, completely degenerated mitochondria with remnants of the cristae was noticed.

Our data suggest that local synthesis of mitochondrial proteins is relevant for their morphology since the mitochondria localized in the dendrites and dendritic spines of *Fmr1* KO mice appear

degenerated. The unusual shape and increased electron density of *Fmr1* KO mitochondria can be an effect of enhanced translation of proteins regulated by FMRP. Interestingly, similar abnormal mitochondrial phenotypes involving a change of shape and size, fragmentation of cristae, and accumulation of osmiophilic material have been described in a study of electron micrographs of autopsy brains of Alzheimer's disease patients [23]. Recently, mitochondrial dysfunction was shown to be connected to autism spectrum disorder [1]. Major neurodegenerative diseases such as Alzheimer's disease, Parkinson's disease, or amyotrophic lateral sclerosis have been associated with mitochondrial dysfunction, including abnormal mitochondrial phenotype [24,25].

In this study, we have made a serendipitous discovery of mitochondrial protein biogenesis at the neuronal synapse. Interestingly, mitochondrial proteins not only represent one of the most abundant classes of locally synthesized proteins in response to neuronal stimulation but they are also capable of entering mitochondria followed by the active formation of functional complexes (synopsis). Besides, our study defines mitochondrial biogenesis as a new group of potential therapeutic targets in diseases such as fragile X syndrome, where the synaptic translation is dysregulated.

# Materials and Methods

### Materials

Key reagents are listed in Table EV2.

### Animals

1- to 2-month-old male FVB mice (FVB/NJ, Jackson Laboratories Stock No.: 001800) were used. For the experiments presented in Figs 5 and EV3, 1- to 2-month-old male *Fmr1* KO (FVB.129-*Fmr1*^tm1Rbd^/J, Jackson Laboratories Stock No.: 008909) mice on FVB background and their wild-type (WT) littermates were used. Before the experiment, the animals were kept in the laboratory animal facility under a 12-h light/dark cycle with food and water available *ad libitum*. The animals were treated in accordance with the EU Directive 2010/63/EU for animal experiments.

### Preparation of synaptoneurosomes and stimulation of NMDA receptors

Synaptoneurosomes were prepared as described previously [16,26,27]. Before tissue dissection, Krebs buffer (2.5 mM $CaCl_2$, 1.18 mM $KH_2PO_4$, 118.5 mM NaCl, 24.9 mM $NaHCO_3$, 1.18 mM $MgSO_4$, 3.8 mM $MgCl_2$, and 212.7 mM glucose) was aerated with an aquarium pump for 30 min at 4°C. Next, the pH was lowered to 7.4 using dry ice. The buffer was supplemented with 1× protease inhibitor cocktail cOmplete EDTA-free (Roche) and 60 U/ml RNase Inhibitor (RiboLock, Thermo Fisher Scientific). Animals were euthanized by cervical dislocation; hippocampi and a part of cortex adjacent to the hippocampus were dissected. Tissue from one hemisphere (~50 mg) was homogenized in 1.5 ml Krebs buffer using Dounce homogenizer with 10–12 strokes. All steps were kept ice-cold to prevent the stimulation of synaptoneurosomes. Homogenates were loaded into 20-ml syringe and passed through a series of

presoaked (with Krebs buffer) nylon mesh filters consecutively 100, 60, 30, and 10 μm (Merck Millipore) in cold room to 50-ml polypropylene tube, centrifuged at 1,000 *g* for 15 min at 4°C, and washed, and pellet was resuspended in Krebs buffer with protease and RNase inhibitors. The protocol for *in vitro* stimulation of NMDA receptors on synaptoneurosomes was described before [16,27]. The aliquots of freshly isolated synaptoneurosomes were prewarmed for 5 min at 37°C and stimulated with a pulse of 50 μM NMDA and 10 μM glutamate for 30 s; then, APV (120 μM) was added and synaptoneurosomes were further incubated for indicated time at 37°C. Unstimulated samples, kept on ice with 200 μg/ml cycloheximide, were used as controls.

### Western blot analysis of synaptoneurosome preparations

Equal amounts of protein from homogenate, filtrate, supernatant (cytosol fraction), and synaptoneurosomal fraction were resolved on SDS–PAGE (10%, TGX Stain-Free FastCast Acrylamide Solutions, Bio-Rad). After electrophoresis, proteins in the gel were visualized using Bio-Rad's ImageLab software to verify the equal protein loading. Proteins were transferred to PVDF membranes (pore size 0.45 μm, Immobilon-P, Merck Millipore) using Trans-Blot Turbo Blotting System (Bio-Rad; 170-4155). Membranes were blocked for 1 h at room temperature in 5% non-fat dry milk in PBS-T (PBS with 0.01% Tween-20), followed by overnight incubation at 4°C with primary antibodies (PSD95, GluA1, GluA2, Nlgn3, synaptophysin, tubulin, GFAP, Gapdh, Hsp90, KDM1, and c-Jun) in 5% milk in PBS-T. Blots were washed 3 × 5 min with PBS-T, incubated for 1 h at room temperature with HRP-conjugated secondary antibody (1:10,000 in 5% milk), and washed 3 × 5 min with PBS-T. In the case of MAPK/Phospho-MAPK Family Antibody Sampler Kit, membranes were blocked for 1 h at room temperature in 5% bovine serum albumin (BSA) in PBS-T; primary and secondary antibodies were diluted also in 5% BSA in PBS-T. HRP signal was detected using Amersham ECL Prime Western Blotting Detection Reagent (GE Healthcare) on Amersham Imager 600 using automatic detection settings.

### Protein labeling with ³⁵S-methionine/cysteine and autoradiography

Freshly isolated synaptoneurosomes were incubated with 100 μCi [³⁵S] methionine/cysteine mix (Perkin Elmer) for 10 min at 4°C. Next, synaptoneurosomes were prewarmed for 5 min at 37°C and stimulated (50 μM NMDA and 10 μM glutamate for 30 s, followed by addition of 120 μM APV). Synaptoneurosomes were further incubated at 37°C for indicated time points. A control sample was stimulated and kept on ice. Synaptoneurosomes were subsequently centrifuged (1,000 ×*g*, 10 min at 4°C), washed with ice-cold Krebs buffer with protease and RNase inhibitors, centrifuged again, and further subjected to SDS–PAGE or BN-PAGE followed by autoradiography or Western blot with specific antibodies.

Where indicated, isolated synaptoneurosomes were stimulated in the presence of 50 μg/ml chloramphenicol (CAP), 3 mM puromycin (PURO), 500 μg/ml cycloheximide, 250 μM anisomycin, and 10 μM carbonyl cyanide m-chlorophenyl hydrazone (CCCP) or VOA mixture (containing 1 μM valinomycin, 20 μM oligomycin, and 8 μM antimycin). For control experiments, isolated

synaptoneurosomes were inactivated at 80°C for 5 min or pretreated with EDTA (50 mM) to disrupt polyribosomes, prior to the stimulation and [$^{35}$S] Met/Cys incorporation.

For SDS–PAGE, proteins were precipitated with pyrogallol red-molybdate (0.05 mM pyrogallol red, 0.16 mM sodium molybdate, 1 mM sodium oxalate, and 50 mM succinic acid) for 20 min at room temperature, centrifuged (20,000 ×g for 15 min), resuspended in urea sample buffer (6 M urea, 6% SDS, 125 mM Tris–HCl [pH 6.8], 0.01% bromophenol blue, and 50 mM DTT), and separated on 15% polyacrylamide gels. Separated proteins were fixed using the Coomassie staining solution. Gels were further destained, dried in a vacuum dryer, and incubated with a storage phosphor screen (GE Healthcare). Radioactively labeled proteins were visualized by digital autoradiography (Storm imaging system; GE Healthcare) followed by image processing with ImageQuant software (GE Healthcare).

For analysis of native protein complexes (blue native electrophoresis, BN-PAGE), synaptoneurosomes and isolated mitochondria were resuspended in solubilization buffer (5% [wt/vol] digitonin, 20 mM Tris–HCl, pH 7.4, 50 mM NaCl, 10% [wt/vol] glycerol, 0.5 mM EDTA, and 2 mM PMSF) and incubated for 20 min on ice. Mitochondria from HEK293T cells were isolated as described in Ref. [28].

Samples were centrifuged at 12,000 ×g at 4°C, and the soluble fraction was mixed with loading dye (5% [w/v] Coomassie Brilliant Blue G-250, 100 mM Bis-Tris, and 500 mM ε-amino-n-caproic acid [pH 7.0]). HMW Native Marker Kit (GE Healthcare) was used for protein complexes' weight reference. Samples were separated on a 4–13% gradient gel at 4°C. For LC-MS analysis, separated protein complexes were subjected to Coomassie staining. After destaining, the bands representing protein complexes were excised from a gel and subjected to LC-MS. Radioactively labeled proteins were visualized by autoradiography as described for SDS–PAGE gels. For Western blotting, protein complexes were transferred to polyvinylidene difluoride membranes (PVDF, Millipore) by a semi-dry transfer (250 mA for 2 h) in the blotting buffer (20 mM Tris, 0.15 M glycine, and 10% methanol). Membranes were incubated with specific primary antibodies and detected by an enhanced chemiluminescence detection system using X-ray films (Foton-Bis). Custom rabbit polyclonal antibodies (from Peter Rehling Lab), anti-COX6A (1:2,000; PRAB3283) and anti-NDUFB8 (1:1,000; PRAB3765), and commercial mouse monoclonal antibody, ATP5A (15H4C4, Abcam, 1:500), SDHA (D-4, Santa Cruz, 1:2,000), and UQCRC2 (13G12AF12BB11, Abcam, 1:500), were used.

## Proteomics

Synaptoneurosomes (unstimulated and 20 min stimulated) were frozen at −80°C; 500 μl of buffer (25 mM HEPES, 2% SDS, protease, and phosphatase inhibitors) was added. Samples were heated at 96°C for 3 min, cooled down, and sonicated in Bioruptor® Plus (Diagenode) for 20 cycles 30/30 s at "high"; then, samples were again heated for 3 min at 96°C.

Protein concentration has been determined using Direct Detect Spectrometer (Merck Millipore). Appropriate volumes containing accordingly 75 μg proteins and 10 μg, respectively, for isobaric labeling and label-free experiments, per sample, were moved to 1.5-ml tubes and precipitated using chloroform/methanol protocol.

Samples then were labeled using standard iTRAQ8-plex (SCIEX) and TMT10-plex (Thermo Fisher) protocol according to the manufacturer's recommendations. Label-free samples were dissolved in 100 μl of 100 mM ammonium bicarbonate buffer, reduced in 100 mM DTT for 30 min at 57°C, alkylated in 55 mM iodoacetamide for 40 min at RT in the dark, and digested overnight with 10 ng/ml trypsin (V5280, Promega) at 37°C. Finally, to stop digestion trifluoroacetic acid was added at a final concentration of 0.1%. The mixture was centrifuged at 4°C, 14,000 g for 20 min, to remove solid remaining.

Samples labeled with TMT and iTRAQ were fractionated prior to LC-MS using strong cation exchange chromatography (SCX). Briefly, 200 μg of samples was injected to PolyLC columns (2.1 × 4.6 × 200 mm, 5 μm, 300 Å pore size) and was fractionated in 60-min linear gradient, from 100% A (5 mM KH$_2$PO$_4$/25% acetonitrile, pH 2.8, or 0.1% formic/30% acetonitrile) to 100% B (5 mM KH$_2$PO$_4$/25% acetonitrile, pH 2.8 + 350 mM KCl or 500 mM ammonium formate/30% acetonitrile).

Proteins in bands excised from a gel were reduced with 100 mM DTT (for 30 min at 57°C), alkylated with 0.5 M iodoacetamide (45 min in a dark at room temperature), and digested overnight at 37°C with 10 ng/μl of trypsin in 25 mM NH$_4$HCO$_3$ (sequencing Grade Modified Trypsin Promega V5111) by adding the enzyme directly to the reaction mixture. Peptides were eluted with 2% acetonitrile in the presence of 0.1% TFA. The resulting peptide mixtures were applied to RP-18 precolumn (Waters, Milford, MA) using water containing 0.1% TFA as a mobile phase and then transferred to a nano-HPLC RP-18 column (internal diameter 75 μM, Waters, Milford MA) using ACN gradient (0–35% ACN in 180 min) in the presence of 0.1% FA at a flow rate of 250 nl/min. The column outlet was coupled directly to the ion source of Orbitrap Velos mass spectrometer (Thermo Electron Corp., San Jose, CA) working in the regime of data-dependent MS to MS/MS switch, and data were acquired in the $m/z$ range of 300–2,000.

MS analysis was performed by LC-MS in the Laboratory of Mass Spectrometry (IBB PAS, Warsaw) using a NanoAcquity UPLC System (Waters) coupled to an Q Exactive Orbitrap Mass Spectrometer (Thermo Fisher Scientific). The mass spectrometer was operated in the data-dependent MS2 mode, and data were acquired in the $m/z$ range of 100–2,000. Peptides were separated by a 180-min linear gradient of 95% solution A (0.1% formic acid in water) to 45% solution B (acetonitrile and 0.1% formic acid). The measurement of each sample was preceded by three washing runs to avoid cross-contamination. Data were analyzed with the MaxQuant (versions 1.5.6.5 and 1.5.7.4) platform using mode match between runs [29]. The mouse proteome database from UniProt was used (downloaded at 2017.01.20). Modifications were set for methionine oxidation, carbamidomethyl or methylthio on cysteines, and phospho (STY). Label-free quantification (LFQ) intensity values were calculated using the MaxLFQ algorithm [30]. Samples labeled with TMT or iTRAQ were searched against the same database with a 0.001 Da error on isobaric tag. Results (LFQ, iTRAQ, and TMT) were analyzed using Scaffold 4 platform (Proteome Software) and merge in a one session file.

## High-resolution respirometry

O$_2$ consumption rates in synaptoneurosomes were measured polarographically using a high-resolution respirometer (Oroboros

Oxygraph-O2K). Freshly isolated synaptoneurosomes (~0.4 mg/ml) were pelleted at 1,000 $g$ for 10 min at 4°C and resuspended in mitochondria respiration medium MIR05 (0.5 mM EGTA, 3 mM MgCl$_2$, 60 mM lactobionic acid, 20 mM taurine, 10 mM KH$_2$PO$_4$, 110 mM sucrose, BSA 0.1%, 20 mM HEPES/KOH, pH 7.1). The density of synaptoneurosomes was approximated by OD at 600 nm, and the volume corresponding to 150 µg per respirator chamber was loaded. Synaptoneurosomes were permeabilized in the chamber by digitonin (0.005%, wt/vol) for 20 min. HEK293T cells were harvested by trypsinization, counted, and resuspended in the MIR05 medium. 2 mln of cells were loaded per respirator chamber and permeabilized by saponin (15 µg/ml) for 20 min. Respiration was measured at 37°C. Malate (0.5 mM) and pyruvate (10 mM) were added to the respirator chambers to support electron entry through complex I followed by titration of CCCP to a final concentration of 2 µM. After inhibition of complex I by rotenone (0.5 µM), succinate (10 mM) was added to support electron entry through complex II. Measurement was concluded with sodium azide (50 mM) to inhibit complex IV. Data recording was performed using Oxygraph-2k and analyzed with DatLab 7 software (Oroboros Instruments). The protein content of synaptoneurosomal and HEK293T samples was measured using Pierce BCA Protein Assay Kit.

### Polyribosome profiling

Polyribosomal profiling of synaptoneurosomes was performed as described previously [16,27]. Synaptoneurosomal samples (unstimulated or NMDA-R-stimulated for 20 min) were centrifuged at 1,000 $g$ for 15 min at 4°C, and the pellet was lysed in 1 ml of lysis buffer [20 mM Tris–HCl pH 7.4, 2 mM DTT, 125 mM NaCl, 10 mM MgCl$_2$, 200 µg/ml cycloheximide, 120 U/ml RNase inhibitor, 1× protease inhibitor cocktail, and 1.5% IGEPAL CA-630 (Sigma Aldrich)]. After centrifugation at 20,000 $g$ for 15 min at 4°C, the supernatant was collected. For RNA-seq experiment, 0.25 vol. of the supernatant was retained as "total" fraction and the remaining 0.75 vol. was loaded on top of a 10–50% linear density sucrose gradient and was centrifuged at 239,316 $g$ for 2 h at 4°C in an SW41 rotor (Beckman Coulter). In the case of polyribosomal profiling using qRT–PCR, whole supernatant was loaded on the gradient. After ultracentrifugation, each gradient was separated and collected into 24 fractions (0.5 ml each) using the BR-188 Density Gradient Fractionation System (Brandel) setup using following parameters: pump speed, 1.5 ml/min; chart speed, 150 cm/h; sensitivity, 0.2; peak separator, off; and noise filter, 1.5. Simultaneously, a continuous absorbance profile of each gradient at 254 nm was graphed. Basing on this profile, fractions were combined into five pools (depicted in Fig 2): (I) monosome, (II) and (III) representing light polysomes, and (IV) and (V) corresponding to heavy polysomes that are engaged in active translation. Obtained fractions were used for RNA isolation and qRT–PCR analysis. For RNA-seq experiment, light and heavy polyribosomal fractions without RNA granules (II, III, IV, and first 1 ml of fraction V) were used (depicted in Fig 2).

### RNA isolation, library preparation, and RNA sequencing

Synaptoneurosomes isolated from cortices and hippocampi of 6 mice were divided into 4 aliquots containing an equal amount of material. One aliquot was saved as a "total" synaptoneurosomal

fraction from which total RNA was extracted with phenol/chloroform mixture, and another was NMDA-R-stimulated for 20 min. Triplicates of ribo-depleted total RNA isolated from synaptoneurosomes ("total") as well as RNA from the NMDA-R-stimulated synaptoneurosome polysome fractions were used to prepare strand-specific libraries (dUTP RNA) [31].

Polysomal fractions were isolated as described above, pooled, and precipitated with 3 volumes of ethanol. Precipitates were digested with proteinase K, and total RNA was extracted with phenol/chloroform mixture. Phase separation was performed using Phase Lock Gel Heavy 2-ml Tubes (5Prime). DNA contamination from 2 µg of nucleic acids was removed by 2 U of TURBO DNase (AM2238, Ambion) in 20 µl of the supplied buffer at 37°C for 30 min. RNA was extracted with phenol/chloroform, precipitated with ethanol, and resuspended in RNase-free water. Concentration was measured with NanoDrop 2000 Spectrophotometer (Thermo Fisher Scientific). Prior to library preparation, to provide an internal performance control for further steps, 1.75 µg of RNA was mixed with 3.51 µl of 1:99 diluted ERCC RNA Spike-In Control Mix 1 (Ambion). Subsequently, rRNA was depleted using the Ribo-Zero Gold rRNA Removal Kit (Human/Mouse/Rat, Illumina) according to the manufacturer's protocol. Fragmentation and first-strand cDNA synthesis were performed as in TruSeq RNA Library Prep Kit v2 protocol (Illumina, RS-122-2001, instruction number 15026495 Rev. D), using SuperScript III Reverse Transcriptase (Thermo Fisher Scientific). For second-strand synthesis, reaction mixtures were supplemented with 1 µl of 5× first-strand synthesis buffer, 15 µl 5× second-strand synthesis buffer (Thermo Fisher Scientific), 0.45 µl 50 mM MgCl, 1 µl 100 mM DTT, 2 µl of 10 mM dUNTP Mix (dATP, dGTP, dCTP, dUTP, 10 mM each, Thermo Fisher Scientific), water to 57 µl, 5 U *E. coli* DNA ligase (NEB), 20 U *E. coli* DNA polymerase I (NEB), and 1 U RNase H (Thermo Fisher Scientific), and incubated at 16°C for 2 h. Further steps, purification, end repair, A-tailing, and adapter ligation, were performed as described in TruSeq kit protocol with one modification: The first purification eluate was not decanted from the magnetic beads, and subsequent steps were performed with the beads in solution. Instead of a new portion of magnetic beads, an equal volume of 20% PEG 8000 in 2.5 M NaCl was added and the DNA bound to the beads already present in the mixture. After the second cleanup procedure after adapter ligation, the supernatant was separated from the beads and treated with USER Enzyme (NEB) in 1× UDG Reaction Buffer (NEB) at 37°C for 30 min. The digestion step ensures that the second strand synthesized with dUTP instead of dTTP is removed from cDNA, resulting in strand-specific libraries. The product was amplified using 1 U of Phusion High-Fidelity DNA Polymerase (Thermo Fisher Scientific) in 1× HF Buffer supplemented with 0.2 mM dNTP Mix, and the following primers:

PP1 (5′-AATGATACGGCGACCACCGAGATCTACACTCTTTCCC TACACGA-3′), and PP2 (5′-CAAGCAGAAGACGGCATACGAGAT-3′). TruSeq kit protocol temperature scheme with 12 amplification cycles and subsequent purification procedure was applied. Enriched library quality was verified using 2100 Bioanalyzer and High Sensitivity DNA Kit (Agilent). The libraries' concentration was estimated by qPCR means with KAPA Universal Library Quantification Kit (Kapa Biosystems), according to the supplied protocol. These libraries were subsequently sequenced using an Illumina HiSeq sequencing platform to the average number of 20 million reads per sample in 100-nt pair-end mode.

Reads were mapped to the mm10 mouse reference genome (GRCm38 primary assembly) using STAR short read aligner with default settings (version STAR_2.5.2) [32] yielding an average of 62% (polysomal fraction) and 85% (Total RNA sample) of uniquely mapped reads. The quality control, read processing and filtering, visualization of the results, and counting of reads to the Gencode vM6 basic annotation were performed using custom scripts utilizing elements of the HTSeq, RSeQC, BEDTools, and SAMtools packages [33–35]. Differential expression analysis between the NMDA-R-stimulated polysome and non-stimulated total RNA samples was performed using the DESeq2 Bioconductor R package [36]. The differential expression analysis from the RNA-seq experiment has been correlated with differential MS results.

### RNA isolation from polysomal fractions and qRT–PCR

Before the RNA extraction, external spike-in control mRNA (an *in vitro* transcribed fragment of A. thaliana LSm gene, 10 ng) was added to each of the fractions to control the extraction step. RNA in each fraction was supplemented with linear polyacrylamide (20 µg/ml) and precipitated with 1:10 volume of 3 M sodium acetate, pH 5.2, and 1 volume of isopropanol. After proteinase K digestion, RNA was isolated using phenol/chloroform extraction method. Phase separation was performed using Phase Lock Gel Heavy 2-ml Tubes (5Prime). RNA pellets were dissolved in 30 µl of RNase-free water. Equal volume from each RNA sample (6 µl) was reverse-transcribed using random primers (GeneON; #S300; 200 ng/RT reaction) and SuperScript IV Reverse Transcriptase (Thermo Fisher Scientific). Next, the abundance of studied mRNAs in different polysomal fractions was analyzed by qRT–PCR using LightCycler 480 Probes Master Mix (Roche) in a LightCycler 480 (Roche). The cDNA was diluted 5× with $H_2O$, and 4 µl of each cDNA sample was amplified using a set of custom sequence-specific primers and TaqMan MGB probes in a final reaction volume of 15 µl.

Following TaqMan Gene Expression Assays (Thermo Fisher Scientific) were used: Dnm1 l (dynamin 1-like); Suclg1 (succinate-CoA ligase, alpha subunit); Sod2 (superoxide dismutase 2); Ndufa10 (NADH dehydrogenase (ubiquinone) 1 alpha subcomplex 10); Sept4 (septin 4); Agk (acylglycerol kinase); Pdhb (pyruvate dehydrogenase E1, beta subunit); Rab35 (RAB35, member RAS oncogene family); Slc25a18 (solute carrier family 25 member 18); and AT3G14080 (U6 snRNA-associated Sm-like protein LSm1).

Relative mRNA levels in different fractions were determined using the ΔCt (where Ct is the threshold cycle) relative quantification method and presented as the % of mRNA in each fraction. Values were normalized to LSm spike-in control.

### MitoPlates

Mitochondrial function assays using MitoPlates S-1 were performed as suggested by the manufacturer. Briefly, assay mix was dispensed into all wells of a MitoPlate and the plate was incubated at 37°C for 1 h to allow substrates to fully dissolve. Freshly isolated synaptoneurosomes were pelleted and resuspended in 1× Biolog Mitochondrial Assay Solution (MAS) containing saponin and dispensed into each well of a MitoPlate (final concentration of saponin 30 µg/ml). Color formation at 590 nm was read kinetically

for 2 h on a Multiskan FC Microplate Photometer (Thermo Scientific). The background was corrected for the blank sample, and the average rate between 10 min and 100 min was calculated. The protein content of synaptoneurosomal samples was measured using Bradford method. Synaptoneurosomes isolated from 3 mice per genotype were analyzed. Results are presented as average rate per minute per µg of protein.

### Serial block-face scanning electron microscopy (SBF-SEM)

*Fmr1* KO and WT mice (n = 3 per genotype) were anesthetized by isoflurane inhalation, injected with Morbital (1–2 ml/kg body weight), and after 10 min transcardially perfused with PBS and afterward with cold 2% paraformaldehyde + 2.5% glutaraldehyde solution in PBS. Brains were isolated and further fixed in the same solution overnight at 4°C. Next, brains were cut on vibratome (Leica VT1000 S) into 100-µm slices and central nucleus of the amygdala (CeA) was manually isolated, washed with PBS (5×3 min, room temperature), and stained on ice with freshly prepared 3% potassium hexacyanoferrate(II) in PBS with 4% aqueous osmium tetroxide in proportion 1:1 for 1 h. Next, samples were rinsed with Milli-Q water (5 × 3 min, room temperature) and incubated in freshly prepared and filtered (0.22-µm syringe filters, Millipore) 1% thiocarbohydrazide aqueous solution for 20 min at room temperature, rinsed again with Milli-Q water (5 × 3 min), and placed in 2% osmium tetroxide solution in Milli-Q water for 30 min in room temperature. After washing with Milli-Q water (5 × 3 min, room temperature), samples were incubated overnight at 4°C in 1% uranyl acetate solution in Milli-Q water. Next day, Walton's lead aspartate was prepared; briefly, 0.66% lead nitrate was added to the 0.4% aspartic acid in $ddH_2O$ and incubated at 60°C for 30 min, and pH was adjusted to 5.5 with 0.1 M NaOH. Tissue samples were washed with Milli-Q water (5 × 3 min), incubated in lead aspartate at 60° C for 30 min, washed with Milli-Q water (5 × 3 min at room temperature), and dehydrated using freshly prepared, ice-cold solutions of 25, 50, 75, and 96% anhydrous ethanol, 5 min each and placed in 100% anhydrous ethanol for 10 min. Next, samples were submerged in a mixture of Durcupan ACM resin:anhydrous ethanol 1:3, 1:1, 3:1 (2 h each) and with 100% resin overnight. Next, tissue pieces were flat-embedded and placed in a 60°C oven for 48 h. Samples were trimmed, mounted into pins, and imaged by SBF-SEM Sigma VP 3View electron microscopy (voltage—14 kV; magnification—10,000×; pixel size—5.9 nm; thickness—60 nm). Samples from 3 mice per genotype were analyzed; from each sample, 4 scans were made—each scan consisted of 120 images.

### Citrate synthase activity

The activity of citrate synthase was assessed as previously described [37] with minor modifications. Briefly, freshly isolated synaptoneurosomal samples were diluted in buffer (50 mM potassium phosphate, pH 7.4, 10% glycerol, and 1% Triton X-100) to a final concentration of 4 mg/ml and incubated for 30 min on ice. 200 µg of synaptoneurosomes was added to the reaction mixture of 20 mM Tris–HCl, pH 8, 0.3 mM acetyl-coenzyme A, and 0.1 mM DNTB and was incubated at RT for 2 min. The reaction was initiated by the addition of 0.5 mM oxaloacetate. The reduction of 5,5′-dithiobis-(2-

nitrobenzoic acid) (DTNB) was monitored spectrophotometrically for 3 min at 412 nm. The protein content of synaptoneurosomal samples was measured using Pierce BCA Protein Assay Kit. Citrate synthase activity is presented as nanomoles per minute and mg of protein [nM/(min*mg protein)].

### Determination of mitochondrial copy number (mitochondrial DNA)

DNA was isolated from synaptoneurosomal samples. SN pellets obtained from one hemisphere (hippocampus and the adjacent part of the cortex) were lysed in 600 µl lysis buffer (10 mM Tris–HCl, pH = 8, 10 mM NaCl, 1 mM EDTA, and 0.5% SDS). Next, 20 µl of proteinase K (Roche, 19 mg/ml) was added and lysates were incubated for 3 h at 55°C and vortexed, and the non-soluble fraction was pelleted by centrifugation at 8,000 $g$ for 15 min. Nucleic acids were extracted using phenol:chloroform:isoamyl alcohol (25:24:1), followed by centrifugation for 15 min at 8,000 $g$. The supernatant was mixed with 1 vol. of chloroform, and after centrifugation for 15 min at 8,000 $g$, nucleic acids were precipitated with 0.1 vol of 3M sodium acetate and 1 vol. of isopropanol at −20°C for 5 min. After centrifugation for 30 min at 12,000 $g$ at 4°C, pellets were washed twice with ice-cold 70% EtOH, air-dried, and dissolved in DNase and RNase-free water. The ratio of mitochondrial DNA (mtDNA) to 18S rRNA was determined by performing quantitative real-time PCR. qRT–PCRs were performed using SYBR Green Master Mix (Thermo Fisher Scientific) in the LightCycler 480 real-time PCR system (Roche) using the following cycling conditions: 95°C for 10 min, followed by 40 cycles of 95°C for 15 s and 60°C for 1 min. 2 ng of DNA was amplified in 15 µl reaction. Primer pairs targeting selected genes, mitochondrial NADH dehydrogenase 1 (MT-Nd1) and 18S rRNA, were used. Fold changes in expression were determined using the ΔΔCT relative quantification method.

### Quantification and statistical analysis

Unless otherwise noted, statistical analysis was performed using GraphPad Prism 7.0 (GraphPad Software, Inc.). Statistical details of experiments, including the statistical tests used and the value of $n$, are noted in figure legends.

Proteomics data were statistically analyzed with the Scaffold 4 (Proteome Software) platform. Groups were compared with the Mann–Whitney test ($P < 0.05$, Benjamini–Hochberg correction). Unstimulated synaptoneurosomes were used as a reference sample. Positive fold change value suggests an increase in the intensity of a given protein as a result of the stimulation.

Interactome was prepared and visually edited with Cytoscape 3.6.0 (3) platform, using the String application.

For differential analysis of the RNA-seq, count data method [36] has been used. The DESeq2 statistical model applies empirical Bayes shrinkage estimation for dispersions and fold change values and Wald test for statistical significance testing. Hits were defined as significantly deregulated when adjusted $P < 0.01$.

Statistical tests used to analyze the data are indicated in the figure legends. Data that were analyzed with parametric tests followed a Gaussian distribution according to the Shapiro–Wilk normality test ($P > 0.05$). Data are presented as means ± SEM.

## Data and software availability

The RNA sequencing data discussed in this publication have been deposited in NCBI's Gene Expression Omnibus [38] and are accessible through GEO Series accession number GSE122724 (https://www.ncbi.nlm.nih.gov/geo/query/acc.cgi?acc = GSE122724).

The proteomic data discussed in this publication have been deposited in PRIDE Archive px-submissions https://www.ebi.ac.uk/pride/archive/projects/PXD012746 and https://www.ebi.ac.uk/pride/archive/projects/PXD012707.

**Expanded View** for this article is available online.

### Acknowledgements
This work was mainly supported by NCN grant SONATA BIS 2014/14/E/NZ3/00375 for MDz; the work in A.C. laboratory was funded by National Science Centre grants OPUS6 2013/11/B/NZ3/00974, OPUS10 2015/19/B/NZ3/03272, ministerial funds for science within Ideas Plus program 000263 in 2014-2017 and "Regenerative Mechanisms for Health" project MAB/2017/2 carried out within the International Research Agendas programme of the Foundation for Polish Science co-financed by the European Union under the European Regional Development Fund. We thank Ben Hur Mussulini for the discussion of experimental procedures. The equipment used was sponsored in part by the Centre for Preclinical Research and Technology (CePT), a project co-sponsored by the European Regional Development Fund and Innovative Economy, The National Cohesion Strategy of Poland.

### Author contributions
MD, AD, and AC designed the research. BK, MW, PS, JM, PK, and MWi performed the research. BK, MW, PS, JM, TMK, and PK analyzed the data. BK, DC, and PS prepared the figures. DC designed proteomics experiments, run MS measurements, analyzed data, and performed bioinformatics analysis and data interpretation. EK and MD provided the resources. MDz wrote the paper. BK prepared Star Methods. All authors reviewed and edited the manuscript.

### Conflict of interest
The authors declare that they have no conflict of interest.

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
