## [Review Process File · EMBO Reports]

Mitochondrial protein biogenesis in the synapse is supported by local translation

Bozena Kuzniewska, Dominik Cysewski, Michal Wasilewski, Paulina Sakowska, Jacek Milek, Tomasz Kulinski, Maciej Winiarski, Pawel Kozielowicz, Ewelina Knapska, Michał Dadlez, Agnieszka Chacinska, Andrzej Dziembowski, and Magdalena Dziembowska

DOI: [10.15252/embr.201948882](https://doi.org/10.15252/embr.201948882)

Corresponding author(s): *Magdalena Dziembowska (m.dziembowska@cent.uw.edu.pl)*

Review Timeline:

Submission Date:	30th Sep 19
Editorial Decision:	25th Oct 19
Revision Received:	25th Feb 20
Editorial Decision:	9th Apr 20
Revision Received:	21st Apr 20
Accepted:	15th May 20

Editor: Deniz Senyilmaz Tiebe

Transaction Report:

Dear Dr. Dziembowska,

Thank you for the submission of your research manuscript to our journal, which was now seen by two referees, whose reports are copied below.

As you can see, the referees express interest in the analysis reporting that locally translated mitochondrial proteins are incorporated into synaptic mitochondria. However, they also raise a number of concerns that need to be addressed to consider publication here. For example, referee #1 requests demonstration of functional relevance of the findings by showing that interfering with the local protein synthesis would lead to mitochondrial dysfunction following stimulation.

Should you be able to address all criticisms in full, we could consider a revised manuscript. I do realize that addressing all the referees' criticisms will require a lot of additional time and effort and be technically challenging. I would therefore understand if you wish to publish the manuscript rapidly and without any significant changes elsewhere, in which case please let us know so we can withdraw it from our system.

If you decide to thoroughly revise the manuscript for EMBO Reports, please address all referee concerns in a complete point-by-point response. Acceptance of the manuscript will depend on a positive outcome of a second round of review. It is EMBO reports policy to allow a single round of revision only and acceptance or rejection of the manuscript will therefore depend on the completeness of your responses included in the next, final version of the manuscript.

Supplementary/additional data: The Expanded View format, which will be displayed in the main HTML of the paper in a collapsible format, has replaced the Supplementary information. You can submit up to 5 images as Expanded View. Please follow the nomenclature Figure EV1, Figure EV2 etc. The figure legend for these should be included in the main manuscript document file in a section called Expanded View Figure Legends after the main Figure Legends section. Additional Supplementary material should be supplied as a single pdf labeled Appendix. The Appendix includes a table of content on the first page with page numbers, all figures and their legends. Please follow the nomenclature Appendix Figure Sx throughout the text and also label the figures according to this nomenclature. For more details please refer to our guide to authors.

2) individual production quality figure files as .eps, .tif, .jpg (one file per figure).

3) a .docx formatted letter INCLUDING the reviewers' reports and your detailed point-by-point responses to their comments. As part of the EMBO Press transparent editorial process, the point-by-point response is part of the Review Process File (RPF), which will be published alongside your paper. For more details on our Transparent Editorial Process, please visit our website: <https://www.embopress.org/page/journal/14693178/authorguide#transparentprocess>

4) a complete author checklist, which you can download from our author guidelines (<<http://embor.embopress.org/authorguide>>). Please insert information in the checklist that is also reflected in the manuscript. The completed author checklist will also be part of the RPF.

5) Please note that all corresponding authors are required to supply an ORCID ID for their name upon submission of a revised manuscript (<<https://orcid.org/>>). Please find instructions on how to link your ORCID ID to your account in our manuscript tracking system in our Author guidelines (<<http://embor.embopress.org/authorguide>>).

6) We replaced Supplementary Information with Expanded View (EV) Figures and Tables that are collapsible/expandable online. A maximum of 5 EV Figures can be typeset. EV Figures should be cited as 'Figure EV1, Figure EV2' etc... in the text and their respective legends should be included in the main text after the legends of regular figures.

- For the figures that you do NOT wish to display as Expanded View figures, they should be bundled together with their legends in a single PDF file called *Appendix*, which should start with a short Table of Content. Appendix figures should be referred to in the main text as: "Appendix Figure S1, Appendix Figure S2" etc. See detailed instructions regarding expanded view here: <<http://embor.embopress.org/authorguide#expandedview>>.

7) We would also encourage you to include the source data for figure panels that show essential data.

Numerical data should be provided as individual .xls or .csv files (including a tab describing the data). For blots or microscopy, uncropped images should be submitted (using a zip archive if multiple images need to be supplied for one panel). Additional information on source data and instruction on how to label the files are available <<http://embor.embopress.org/authorguide#sourcedata>>.

8) Our journal encourages inclusion of *data citations in the reference list* to directly cite datasets that were re-used and obtained from public databases. Data citations in the article text are distinct from normal bibliographical citations and should directly link to the database records from which the data can be accessed. In the main text, data citations are formatted as follows: "Data ref: Smith et al, 2001" or "Data ref: NCBI Sequence Read Archive PRJNA342805, 2017". In the Reference list, data citations must be labeled with "[DATASET]". A data reference must provide the database name, accession number/identifiers and a resolvable link to the landing page from which the data

can be accessed at the end of the reference. Further instructions are available at <http://embor.embopress.org/authorguide#datacitation>.

9) Before submitting your revision, primary datasets (and computer code, where appropriate) produced in this study need to be deposited in an appropriate public database (see <http://embor.embopress.org/authorguide#dataavailability>).

The accession numbers and database should be listed in a formal "Data Availability" section (placed after Materials & Method) that follows the model below. Please note that the Data Availability Section is restricted to new primary data that are part of this study.

Data availability

10) Regarding data quantification, please ensure to specify the name of the statistical test used to generate error bars and P values, the number (n) of independent experiments underlying each data point (not replicate measures of one sample), and the test used to calculate p-values in each figure legend. Discussion of statistical methodology can be reported in the materials and methods section, but figure legends should contain a basic description of n, P and the test applied. Please note that error bars and statistical comparisons may only be applied to data obtained from at least three independent biological replicates. Please also include scale bars in all microscopy images.

I look forward to seeing a revised version of your manuscript when it is ready. Please let me know if you have questions or comments regarding the revision.

Yours sincerely,

Deniz Senyilmaz Tiebe

Deniz Senyilmaz Tiebe, PhD
Editor
EMBO Reports

Referee #1:

Kuzniewska and colleagues address an interesting question about how heightened energy demands after stimulation are met at the synapse. They report that stimulation of synaptoneurosomes triggers the local translation of mitochondrial proteins and their incorporation into mitochondria. The individual aspects of this finding are not extremely novel: the local synthesis of nuclear-encoded mitochondrial proteins in neurites is well established, as is the stimulation-dependent induction of protein synthesis in synaptoneurosomes following stimulation. The main conceptual advance of this manuscript is thus the demonstration that newly synthesized mitochondrial proteins are getting incorporated in synaptic mitochondria. The impact of this paper could be improved if the authors would show that interference with the local synthesis of mitochondrial proteins leads to energy deficits following stimulation and failure to maintain the membrane potential.

The authors use three different MS approaches to identify proteins that are upregulated in their synaptoneurosomal preps. The experiments in the manuscript are overall well performed, documented and analyzed. However, the main findings are based on metabolic labelling experiments that are insufficiently controlled and the presented results of these experiments do not unambiguously support the authors' interpretation.

Specific concerns:

Figure 1B: the authors present primarily markers for synaptoneurosomes but not the absence of markers for contaminants. It is crucial to establish that their prep does not contain cell bodies/nuclei or glial cells.

Figure 1C: the puromycin control is unconvincing. Despite using an extraordinarily high concentration of puromycin (3mM), the 'incorporation' of 35S-Met/Cys is only partially (~50%) inhibited. This finding makes it difficult to interpret the presented data as incorporation vs. stickiness of the labelled amino acids. The authors should use another protein synthesis inhibitor (PSI), and include the PSI condition in the following experiments (see below).

Figure 2C: the difference in the abundance of mitochondrial proteins among total and upregulated proteins is negligible, challenging the authors' view that local protein production is specifically maintaining functional mitochondria.

Figure 4B,C: because of the high labeling signal even in the presence of puromycin (Fig.1C), it is not possible to claim that the 35S positive bands are indeed representing incorporation of newly synthesized proteins. The authors should include PSI conditions in these experiments.

Figure 4B,C: C should be analyzed and normalized like B (i.e. against ctr.).

General: Does import of mitochondrial proteins really signify mitochondrial biosynthesis? I would suggest changing the wording to more accurately reflect the finding. Alternatively, if the authors want to claim biogenesis, they should show changes mitochondrial mass, etc.

Referee #2:

EMBOR-2019-48882V1

Mitochondria biogenesis in the synapse is supported by local translation.

Bozena Kuzniewska et al.

In this manuscript, the authors use synaptoneurosomes to profile the proteins whose

local translation are increased by stimulation. From this, they demonstrate that there is a pool of mitochondrial proteins that are locally synthesized, that transition into actively translated polysomal fractions, and that are ultimately incorporated into the mitochondrial respiratory chain supercomplexes. The relative abundance of mRNAs encoding mitochondrial proteins in axons, dendrites, and synapses has been demonstrated over the years by several groups; however, this study provides important additional information on the incorporation of specific proteins into the mitochondria, and how this is altered upon stimulation, and as such contributes to our understanding of the importance of local translation to the proper function of the synapse. Overall, it is likely to be of interest to journal readers; however, I have several questions, comments, and additions, that if adequately addressed would strengthen the manuscript and might make it suitable for publication.

Major:

1. The authors have published a prior methods paper on the synaptoneurosomal fractionation that they are using here. In my opinion, neither that paper nor this one adequately validate this preparation. The markers they use show enrichment of pre- and post-synaptic markers, which is important; however, equally important are negative or reduced markers. Are GFAP levels reduced (as a measure of non-synaptosomal astrocytic contribution)? How about nuclear markers such as Histones or Lamin-B1? Cytosolic markers such as Hsp90? Given the fundamental assertion in this manuscript that the data represent evidence of synaptosomal local translation of these mitochondrial proteins, it is absolutely critical that the preparation being used be adequately validated. Also, prior studies have shown that synaptoneurosomal preparations can incorporate radiolabeled cysteine/methionine, and they have also demonstrated that percoll gradient preparations are much more translationally active than filtration preparations as are used here.
2. I think the Discussion of the results in Figure 2 need expansion. It isn't clear to me if there are mRNAs that are increased in the polysomal fraction, but show a paradoxical decrease in protein levels (or vice versa). Do directional changes always correlate? If not, what does mean? Also, there is a failure to discuss mitochondrial proteins that are decreased with stimulation and what that might mean, either biologically or with respect to the methodologies used.
3. The metabolic labeling throughout isn't particularly convincing in terms of increase, but particularly in terms of inhibition. For example, treatment with CCP or VOA to inhibit protein transport into mitochondria doesn't really do much. Figure 4B-C is not particularly convincing. Similarly, the inhibition by puromycin in Figure 1C.

Minor:

1. The manuscript is very clearly written; however there are a few places where the language could benefit from editing prior to publication.
2. For the polyribosomal profiling, how long is the stimulation prior to isolation of the polysomal fractions?
3. I'm not sure I fully appreciate what is being shown in Supplemental Figure S2. Some of the transcripts behave as expected, with a decrease in a lower fraction and a corresponding increase in a higher fraction with stimulation (indicative of an increase in translation and increased protein levels); however, others do not. Do these transcripts fall into different classes in terms of their protein level changes with stimulation?

Point by point response to the Reviewer's comments

We thank the Reviewers for a positive opinion about our paper. We address all issues raised by Reviewers and edited the manuscript to improve its clarity. Below we provide a point by point response to the Reviewer's comments.

Referee #1:

Both, Referee #1 and the Editor requested the demonstration of the functional relevance of the findings by showing that interfering with the local protein synthesis would lead to mitochondrial dysfunction.

The best-described model of the dysregulated local protein synthesis at the synapse is the Fmr1 KO mouse. It is a model of fragile X syndrome, a genetic condition caused by the silencing of the Fmr1 gene. Fmr1 is an RNA binding protein repressing basal translation. Therefore we decided to use Fmr1 KO mice to study the effect of dysregulated local protein synthesis in synapses on mitochondrial physiology. Analysis of synaptic mitochondria in synaptoneurosomes isolated from Fmr1 KO revealed altered activity of OXPHOS complexes. Moreover, synaptic mitochondria in Fmr1 KO mouse brain turned out to have abnormal morphology (visualized by electron microscopy), which proves that, indeed, the local synthesis of mitochondrial proteins has functional relevance. The new data are now presented in Figure 5 and EV3 and discussed in the manuscript on pages 8 and 9.

Specific concerns:

Figure 1B: the author present primarily markers for synaptoneurosomes but not the absence of markers for contaminants. It is crucial to establish that their prep does not contain cell bodies/nuclei or glial cells.

As suggested by the Reviewer, we have additionally validated our synaptoneurosomal preparations and replaced a panel on Figure 1B with a new, revised version. We show enrichment of pre- and postsynaptic markers (Psd95, GluA1, GluA2, Nlgn3, synaptophysin) as well as depletion of cytosolic markers (Gapdh and Hsp90) in synaptoneurosomes as compared to the homogenates. Nuclear markers (Kdm1 and c-Jun) are barely detectable in synaptoneurosomal fraction. Glia marker (Gfap) is present in synaptoneurosomes; however, it is not enriched. The results are described on page 4 of the manuscript.

Figure 1C: the puromycin control is unconvincing. Despite using an extraordinarily high concentration of puromycin (3mM), the 'incorporation' of 35S-Met/Cys is only partially (~50%) inhibited. This finding makes it difficult to interpret the presented data as incorporation vs. stickiness of the labelled amino acids. The authors should use another protein synthesis inhibitor (PSI), and include the PSI condition in the following experiments (see below).

We thank the Reviewer for this comment. As suggested, we have added additional experiments using other protein synthesis inhibitors – cycloheximide and anisomycin to show the inhibition of 35S-Met/Cys incorporation (Fig. EV1C). Moreover, in order to verify the specificity of the labeling and rule out the possibility of the unspecific stickiness of the labeled amino acids to the proteins, we performed additional control experiments. Synaptoneurosomes were “inactivated” at 80°C or

pretreated with EDTA, (which entirely disrupts polyribosomes by causing the dissociation of the large and small ribosomal subunits) before the labeling. To strengthen our data, we also included the experiments in which SNs were incubated with VOA mixture (valinomycin, oligomycin, antimycin) to inhibit protein transport into the mitochondria. In these conditions, 35S-Met/Cys incorporation was completely inhibited. New data has been added to the extended view of Figure 1 (Figure EV1 panel B, C and D) and described on page 4 of the manuscript.

Figure 2C: the difference in the abundance of mitochondrial proteins among total and upregulated proteins is negligible, challenging the authors' view that local protein production is specifically maintaining functional mitochondria.

We thank the Referee for this remark, the Reviewers' question made us aware that the presentation of the data was somehow misleading. We did not intend to suggest that local protein production is exclusively maintaining mitochondria; of course, other synaptic proteins are translated in response to the stimulation. What was shown on former Fig.1F (label free MS) and Fig.2D (iTRAQ8/TMT10) was the percentage share of mitochondrial proteins in the pool of all identified proteins (upper pie-chart) and percentage share of mitochondrial proteins in the pool of upregulated proteins (lower pie-chart). This data are qualitative. The quantitative analysis of each protein level is presented on the volcano plots (Fig.1D and Fig. 2 AB) or in Table 1.

To make our point more clear, we replaced the panels on Figures 1F and 2C with pie-charts showing only the percentage share of mitochondrial proteins in the pool of upregulated proteins. Figure legends were changed accordingly.

Figure 4B,C: because of the high labeling signal even in the presence of puromycin (Fig.1C), it is not possible to claim that the 35S positive bands are indeed representing the incorporation of newly synthesized proteins. The authors should include PSI conditions in these experiments.

As described above, we performed additional controls to verify the specificity of the labeling on SDS-PAGE (Figure EV1). Also, to answer Reviewers' question we performed additional experiments using puromycin treatment to verify the specificity of 35S labeling revealed on the BN-PAGE autoradiography (Fig. 4). A new panel showing the effect of puromycin treatment on the incorporation of newly synthesized proteins into respiratory chain complexes has been added as Figure 4C and described in the manuscript on page 8.

As compared to SDS-PAGE, BN-PAGE autoradiography may present higher levels of apparent labeling of protein complexes due to the unspecific binding of labeled aminoacids to native complexes. We were able to decrease the background by denaturing the complexes with SDS before the transfer to the membrane.

Figure 4B,C: C should be analyzed and normalized like B (i.e. against ctr.).
General: Does import of mitochondrial proteins really signify mitochondrial biosynthesis? I would suggest changing the wording to more accurately reflect the finding. Alternatively, if the authors want to claim biogenesis, they should show changes mitochondrial mass, etc.

As suggested by the Reviewer, the data presented in Figure 4C was recalculated and normalized to unstimulated control as in Figure 4B. Also, additional experiments were performed and added to the

analysis (both using VOA mixture and puromycin). Figure 4C was replaced with a new, revised version.

We agree with the Reviewer's remark that the wording "mitochondrial biogenesis" may be misleading in the context of our findings. Therefore we modified the title to more specifically describe our discovery, that in fact concerns "mitochondrial protein biogenesis". The manuscript was revised accordingly. The term "protein biogenesis" is commonly used as a description of all the processes that a protein is subjected, from the time of its synthesis through transport, modifications to the assembly into the functional units (i.e. used as a title in the recent review of Pfanner et al., 2019).

Referee #2:

Major:

1. The authors have published a prior methods paper on the synaptoneurosomal fractionation that they are using here. In my opinion, neither that paper nor this one adequately validate this preparation. The markers they use show enrichment of pre- and post-synaptic markers, which is important; however, equally important are negative or reduced markers. Are GFAP levels reduced (as a measure of non-synaptosomal astrocytic contribution)? How about nuclear markers such as Histones or Lamin-B1? Cytosolic markers such as Hsp90? Given the fundamental assertion in this manuscript that the data represent evidence of synaptosomal local translation of these mitochondrial proteins, it is absolutely critical that the preparation being used be adequately validated. Also, prior studies have shown that synaptoneurosomal preparations can incorporate radiolabeled cysteine/methionine, and they have also demonstrated that percoll gradient preparations are much more translationally active than filtration preparations as are used here.

As suggested by the Reviewer, we have additionally validated our synaptoneurosomal preparations and replaced a panel on Figure 1B with a new, revised version. We show enrichment of pre- and postsynaptic markers (Psd95, GluA1, GluA2, Nlgn3, synaptophysin) as well as depletion of cytosolic markers (Gapdh and Hsp90) in synaptoneurosomes as compared to the homogenate. Nuclear markers (Kdm1 and c-Jun) are barely detectable in synaptoneurosomal fraction. Glia marker (Gfap) is present in synaptoneurosomes; however, it is not enriched. The results are described on page 4 of the manuscript.

The protocol of SN preparation used in our experiments was developed on the basis of earlier protocols, which proved useful for the study of local protein translation (Scheetz et al., 2000; Mudashetty et al., 2007). We used SN in our previous work to prove activity-dependent local translation of MMP-9 (Dziembowska et al 2012, Janusz et al 2013) or neurologins (Chmielewska et al 2018).

2. I think the Discussion of the results in Figure 2 need expansion. It isn't clear to me if there are mRNAs that are increased in the polysomal fraction, but show a paradoxical decrease in protein levels (or vice versa). Do directional changes always correlate? If not, what does mean? Also, there is a failure to discuss mitochondrial proteins that are decreased with stimulation and what that might mean, either biologically or with respect to the methodologies used.

We thank the Reviewer for this comment. Indeed, the description and visualization of our data were not clear enough, and in the revised version of the manuscript, this section was modified (page 6). In Figure 2F we have added the density curves for transcripts encoding mitochondrial proteins as well as transcripts encoding upregulated proteins detected by the mass spectrometry. Looking globally on our data, there is striking accordance between the transcriptomic and MS results as it is clear that in general, mRNAs encoding proteins that based on MS results are locally translated have high polysome occupancy. Comparing very different datasets obtained with such distinct methods as mass spectrometry and RNA-seq of polyribosomal fractions poses multiple challenges, so in our opinion, the observed consistency of trends is very convincing. Moreover, the engagement of several mRNAs encoding mitochondrial proteins into translation was also verified by RT-qPCR on the polysomal fractions (Fig.EV2).

Proteomic and transcriptomic datasets arise from very different experimental setups and therefore have divergent biases leading to the increased experimental noise. Annotations commonly used in proteomics and transcriptomic analysis, as well as difficulties in normalization, are not fully compatible, hampering a direct comparison of datasets. Moreover, some examples of mRNAs encoding proteins of increased abundance after stimulation which are not enriched in the polysomal fraction and *vice versa* may also have biological explanations. During the stimulation, proteins are not only produced but also degraded. In the *in vitro* system we use, some proteins may have degradation rates exceeding synthesis. In the case of mRNA present in synapses, some of them are repressed at the elongation stage of translation, and after the stimulation, although the translation is activated, there will be no increases in polysome occupancy. Every single case of discrepancy can be studied. However, we believe that such analyses are outside of the scope of this paper, which focuses on the description of the general phenomenon rather than individual proteins. Importantly a good correlation between protein and RNA analysis supports our conclusion that proteins synthesized at the synapses can build mitochondria.

3. The metabolic labeling throughout isn't particularly convincing in terms of increase, but particularly in terms of inhibition. For example, treatment with CCP or VOA to inhibit protein transport into mitochondria doesn't really do much. Figure 4B-C is not particularly convincing. Similarly, the inhibition by puromycin in Figure 1C.

In order to address this comment of the Reviewer, we have added additional experiments using other protein synthesis inhibitors – cycloheximide and anisomycin to show the inhibition of 35S-Met/Cys incorporation (Fig. EV1C). Moreover, in order to verify the specificity of the labeling and rule out the possibility of the unspecific stickiness of the labeled amino acids to the proteins, we performed additional control experiments. Synaptoneurosomes were “inactivated” at 80°C or pretreated with EDTA, (which entirely disrupts polyribosomes by causing the dissociation of the large and small ribosomal subunits) before the labeling. To strengthen our data, we also included the experiments in which SNs were incubated with VOA mixture (valinomycin, oligomycin, antimycin) to inhibit protein transport into the mitochondria. In these conditions, 35S-Met/Cys incorporation was completely inhibited. New data has been added to the extended view of Figure 1 (Figure EV1 panel B, C and D) and described on page 4 of the manuscript.

We also performed additional experiments using VOA and puromycin treatment to verify the specificity of 35S labeling revealed on the BN-PAGE autoradiography (Fig. 4). A new panel showing

the effect of puromycin treatment on the incorporation of newly synthesized proteins into respiratory chain complexes has been added as Figure 4C and described in the manuscript on page 8. As compared to SDS-PAGE, BN-PAGE autoradiography may present higher levels of apparent labeling of protein complexes due to the unspecific binding of labeled aminoacids to native complexes. We were able to decrease the background by denaturing the complexes with SDS before the transfer to the membrane.

As described above, we provide a number of controls to ensure the specificity of the labeling. Below we present the data for CCCP, where we observed dose-response inhibition of 35S incorporation, for both BN-PAGE (upper panel) and SDS-PAGE (lower panel). For further experiments, we decided to use CCCP in the concentration of 10 μM, as it was reducing the incorporation of 35S-Met/Cys to the control level.

Minor:

1. The manuscript is very clearly written; however there are a few places where the language could benefit from editing prior to publication.

The professional text editor corrected the language in the manuscript.

2. For the polyribosomal profiling, how long is the stimulation prior to isolation of the polysomal fractions?

We apologize for not providing this information, it was lacking due to mistake. It was corrected in the revised version of the manuscript on pages 12, 13, 19, 20 and 22. For mass spectrometry analysis, RNA-Seq and polyribosomal profiling, the samples were collected 20 minutes after the stimulation.

3. I'm not sure I fully appreciate what is being shown in Supplemental Figure S2. Some of the transcripts behave as expected, with a decrease in a lower fraction and a corresponding increase in a higher fraction with stimulation (indicative of an increase in translation and increased protein levels); however, others do not. Do these transcripts fall into different classes in terms of their protein level changes with stimulation?

In order to address this comment, we revised Figure S2 (which is now Figure EV2) and changed graph type into the stacked-bars, which, as we hope, will be a more precise presentation of the data. Also, we modified the text describing this figure, as it was inadequately explained in the manuscript. All of the transcripts presented in figure EV2 fall into the same category – their abundance increases in heavy-polysomal fractions upon the stimulation at the expense of monosome/light polysomal fractions. At the same time, their protein levels are increased in response to the stimulation (as analyzed by mass spectrometry).

We appreciate your careful evaluation of our work that helped us to improve the quality of the paper. We hope that this revision meets with your approval. We have included the revised manuscript version that highlights the changes from the original submission (in blue).

Dear Dr. Dziembowska,

Thank you for submitting your revised manuscript. It has now been seen by both of the original referees.

As you can see, the referees find that the study is significantly improved during revision and recommend publication. Before I can accept the manuscript, I need you to address some minor points below:

- We noticed that Ewelina Knapska is currently missing from the Author Contributions section.
- We noted that Figure 5A is not called out in the text.
- We realized that currently there are 2 EV tables. Table EV1 should be made a Data Set and relabelled accordingly. A legend should be directly added to the file. Table EV2 should be made Table EV1.
- Please make both the proteomics data deposited to PRIDE and the RNAseq data deposited on GEO accessible.
- We noted that there is a 'Conclusions' section in the manuscript, which is not compliant with the journal format. Please remove the title and add the text to the 'Results & Discussion' section.
- Our production/data editors have asked you to clarify several points in the figure legends (see attached document). Please incorporate these changes in the attached word document and return it with track changes activated.

Thank you again for giving us to consider your manuscript for EMBO Reports, I look forward to your minor revision.

Kind regards,

Deniz Senyilmaz Tiebe

--

Deniz Senyilmaz Tiebe, PhD
Editor
EMBO Reports

Referee #1:

The authors have answered the original comments satisfactorily and the newly included controls have significantly enhanced the quality of the manuscript. In the newly added figure 2F, I would recommend to color code only the dots that are significant (red) and to restrict the density analysis only to those significant mRNAs.

Referee #2:

EMBOR-2019-48882V2

Mitochondria protein biogenesis in the synapse is supported by local translation.

Bozena Kuzniewska et al.

In this manuscript, the authors use synaptoneurosome preparations to profile the proteins whose local translation are increased by stimulation. From this, they demonstrate that there is a pool of mitochondrial proteins that are locally synthesized, that transition into actively translated polysomal fractions, and that are ultimately incorporated into the mitochondrial respiratory chain supercomplexes. The relative abundance of mRNAs encoding mitochondrial proteins in axons, dendrites, and synapses has been demonstrated over the years by several groups; however, this study provides important additional information on the incorporation of specific proteins into the mitochondria, and how this is altered upon stimulation, and as such contributes to our understanding of the importance of local translation to the proper function of the synapse. Overall, it is likely to be of interest to journal readers. This is a revision of a previously submitted manuscript. The authors have been very responsive to criticisms in the original submission, and the revised manuscript is substantially improved. In particular, inclusion of additional controls for the synaptoneurosomal preparations, clarification of treatments conditions, inclusion of additional protein synthesis inhibitors, and new experiments addressing the functional significance of altered protein synthesis all strengthen the manuscript. Given this, I believe that the manuscript is now suitable for publication and will be of significant interest to readers.

February 25th, 2020
EMBO Reports

Dear Dr. Tiebe

Thank you for giving us the opportunity to resubmit our manuscript entitled "Mitochondrial protein biogenesis in the synapse is supported by local translation". We thank the Reviewers for their comments and insightful questions. In order to meet the Reviewers' criticism, we have performed additional experiments and we hope that you will find our manuscript substantially improved. Notably, we have addressed all the issues raised by Reviewers.

In brief, we aimed at showing the functional relevance of the local biogenesis of synaptic mitochondria. To this end, we decided to include new exciting data obtained on the *Fmr1* KO mice, a model of dysregulated synaptic translation. Analysis of synaptic mitochondria in synaptoneurosomes isolated from *Fmr1* KO revealed altered activity of OXPHOS complexes. Moreover, synaptic mitochondria in *Fmr1* KO mouse brain turned out to have abnormal morphology (visualized by electron microscopy), which proves that, indeed, the local synthesis of mitochondrial proteins has functional relevance. Finally, we have introduced all controls requested by the Reviewers.

Since the revised version of the manuscript contains new data and has been significantly restructured and rewritten, we hope that you will consider this new revised version favorably.

Sincerely yours,

Magdalena Dziembowska

Dear Magda,

Thank you for sending your revised manuscript. I have now looked at everything and all looks fine. Therefore I am very pleased to accept your manuscript for publication in EMBO Reports.

Congratulations on a nice work!

Kind regards,

Deniz

--

Deniz Senyilmaz Tiebe, PhD

Editor

EMBO Reports

Corresponding Author Name: Magdalena Dziembowska

Manuscript Number: EMBOR-2019-48882V2